# Mastocytosis-derived extracellular vesicles deliver miR-23a and miR-30a into pre-osteoblasts and prevent osteoblastogenesis and bone formation

Do-Kyun Kim[1,2,8], Geethani Bandara[1], Young-Eun Cho[3,4], Hirsh D. Komarow[1], Danielle R. Donahue [5], Baktiar Karim[6], Moon-Chang Baek [7], Ho Min Kim [2], Dean D. Metcalfe [1,9] & Ana Olivera [1,9✉]

Osteoporosis and other manifestations of bone disease are frequent in patients with systemic mastocytosis (SM) in association with the presence of mast cell infiltrates in bone marrow, although the mechanisms behind bone disease remain poorly understood. We find that extracellular vesicles (EVs) released by neoplastic mast cells and present in the serum of patients with SM (SM-EVs) block osteoblast differentiation and mineralization in culture, and when injected into mice diminish the expression of osteoblast markers, and trabecular bone volume and microarchitecture. We demonstrate that miRNA-30a and miRNA-23a, increased in SM-EVs and neoplastic mast cell-derived EVs, attenuate osteoblast maturation by suppressing expression of RUNX2 and SMAD1/5, essential drivers of osteogenesis. Thus, SM-EVs carry and deliver miRNAs that epigenetically interfere with bone formation and can contribute to bone mass reduction in SM. These findings also suggest possibilities for novel approaches to the management of bone disease in mast cell proliferative disorders.

[1] Mast Cell Biology Section, Laboratory of Allergic Diseases, National Institute of Allergy and Infectious Diseases, National Institutes of Health (NIH), Bethesda, MD, USA. [2] Center for Biomolecular and Cellular Structure, Institute for Basic Science (IBS), Daejeon, Republic of Korea. [3] Section of Molecular Pharmacology and Toxicology, Laboratory of Membrane Biochemistry and Biophysics, National Institute on Alcohol Abuse and Alcoholism, NIH, Bethesda, MD, USA. [4] Department of Food and Nutrition, Andong National University, Andong, Kyungpook, Republic of Korea. [5] Mouse Imaging Facility, National Institute of Neurological Disorders and Stroke, NIH, Bethesda, MD, USA. [6] Molecular Histopathology Laboratory, Leidos Biomedical Research, Inc., Frederick National Laboratory for Cancer Research, National Cancer Institute, NIH, Frederick, MD, USA. [7] Department of Molecular Medicine, CMRI, Exosome Convergence Research Center (ECRC), School of Medicine, Kyungpook National University, Daegu, Republic of Korea. [8] Present address: Korea Zoonosis Research Institute, Jeonbuk National University, Iksan, Jeonbuk, Republic of Korea. [9] These authors contributed equally: Dean D. Metcalfe, Ana Olivera. ✉email: ana.olivera@nih.gov

Systemic mastocytosis (SM) is a hematological disease caused by abnormal accumulation of mast cells in bone marrow (BM), skin, and other tissues, associated with somatic variants of the receptor for stem cell factor, KIT[1]. Osteoporosis of the long bones and the spine in patients with SM is one of the more debilitating aspects of this disease[2–5]. Although clinical studies in patients with SM with bone disease suggest high bone turnover with an imbalance between bone formation and bone resorption[6–8], and mast cell-derived products are believed to contribute to such imbalances[6,8], the pathogenesis of osteoporosis in SM and the mechanisms involved remain poorly understood.

In osteoporosis, bone density is reduced as a result of an uncoupling between bone formation and bone loss, causing microarchitecture deterioration, bone fragility, and porosity. The balance between bone resorption and bone formation is tightly regulated by several factors secreted systemically or in the local microenvironment that link these two cellular processes in a controlled coupling. Osteoblasts, osteoclasts, stromal BM cells, and bone hematopoietic cells are considered main contributors to this milieu of factors[9,10]. However, mast cell numbers in the bone have been found in association with osteoporosis in rodents[11,12] and in humans, not only in patients with mastocytosis but in menopausal women and other cases of osteoporosis where most common secondary causes were excluded[13,14]. This led to the idea that mast cells may contribute to loss of bone homeostasis. Indeed, mast cell-related products including histamine may promote bone resorption[4,15] and inhibit osteoblast differentiation[16]. Furthermore, in SM, mast cells produce receptor activator of nuclear factor-κB ligand (RANKL) and the decoy receptor for RANKL, osteoprotegerin (OPG), suggesting that neoplastic mast cells regulate bone turnover since RANKL promotes while OPG prevents osteoclastogenesis[6].

Growing evidence indicates that in addition to soluble factors, extracellular vesicles (EVs) released by osteoclasts, osteoblasts, and BM stromal cells, acting as vehicles for the crosstalk between these cells, may critically regulate osteoclastogenesis as well as osteoblast differentiation and mineralization[17–20]. EVs, including exosomes and microvesicles, are released by all cell types and especially neoplastic cells; and contain cell-type-specific and selectively packaged proteins, mRNAs and small RNAs that are transported stably in circulation and taken up by recipient cells in a process believed to be nonrandom. Delivery of these bioactive molecules may cause functional changes in recipient cells[21]. Recently, we demonstrated that patients with SM have increased concentrations of small EVs in circulation with characteristic mast cell markers that have the potential of altering the phenotype of non-hematopoietic cells[22]. Because mast cells also accumulate in the BM of patients with SM, we thus investigated the possibility that mast cell-derived EVs from these patients could alter normal bone coupling.

Herein, we demonstrate that SM-derived EVs (SM-EVs) with a mast cell signature[22] and EVs derived from neoplastic mast cells contain fate-modifying miRNAs that are taken up by pre-osteoblastic cells and prevent their differentiation into osteoblasts in vitro and in vivo. Specifically, miRNA-30a and miRNA-23a caused repression of the master regulator of osteogenesis, runt-related transcription factor 2 (RUNX2), and SMAD proteins, which transduce signals from bone morphogenic proteins (BMPs), also critical drivers of osteoblast differentiation. Our study brings forward the concept that increased numbers of neoplastic mast cells in patients with SM, via the production of EVs, can cause perturbations in the pool of miRNAs that regulate normal bone homeostasis. This underscores the potential of SM-EVs to contribute to various aspects of pathogenesis in SM by a broad spectrum of mechanisms including transfer of oncogenic proteins, as we previously reported in hepatic stellate cells[22] or, as

shown here, causing epigenetic alterations in targets cells. To our knowledge this is the first report identifying neoplastic mast-cell-derived EVs and the miRNAs they carry and deliver, in the negative regulation of osteogenesis. These observations add to the understanding of osteoporosis in SM and reinforces the importance of EVs in mast-cell-related diseases.

## Results

**SM-EVs inhibit osteoblast differentiation and mineralization.** As demonstrated in our previous report[22], the concentrations of EVs in the serum of patients with SM are increased compared to healthy volunteers (Fig. 1a) in correlation with tryptase levels, a surrogate marker of mast cell burden (see "Methods" and Supplementary Table 1, Group 1, for a more detailed description of the patients, divided into those with tryptase levels lower or higher than the median value in his cohort). Gender did not appear to be a factor contributing the concentration of EVs in circulation (Supplementary Fig. 1). The average size of these EVs, determined by nanoparticle tracking analysis (NTA) (Fig. 1b) and confirmed by EM imaging[22], was <90 nm, consistent with the size of small EVs (50–150 nm). The size distribution was homogeneous, with 90% of the EVs in the range of 30–200 nm (Fig. 1b). These SM-EVs, similar to HV-EVs, contained typical EV markers such as CD9 and ALIX, but unlike HV-EVs, exhibited characteristic mast cell markers including the Mas-related G-protein-coupled receptor X2 (Fig. 1c) and other mast cell markers such as tryptase, high affinity IgE receptors, and KIT[22]. Since bone homeostasis is altered in patients with SM, presumably related to the excess accumulation of mast cells within the marrow and other tissues, we explored the potential contributions of these EVs with mast cell markers in osteoblastogenesis.

We first used an osteoblastic cell line derived from human cord blood, hFOB1.19 cells[23,24]. As expected, hFOB1.19 plated at 70–80% confluency and cultured at 39.5 °C for 15 days (see "Methods"), progressively differentiated into mature osteoblasts. This was evidenced by the increased production of alkaline phosphatase (ALP), which reached a plateau by 6–15 days (Fig. 2a, open circles); the change into spindled-shaped cells and substantial deposition of extracellular calcium phosphate nodules visualized with alizarin red (Fig. 2b, upper panel, left figure); and the increase in expression of RUNX2 and bone matrix proteins produced by osteoblasts such as osteopontin (OPN) and type I collagen (COL1) (Fig. 2c). Treatment of the cultures with SM-EVs (with every change of media) starting at the time of plating, 3 or 6 days after plating (Fig. 2a), resulted in substantial reductions in ALP activity by these cultures, while treatments with healthy volunteer (HV)-EVs had no effect. Similarly, SM-EVs reduced the deposition of calcium by >50% compared to untreated osteoblastic cultures or cultures treated with HV-derived EVs, as evidenced by the apparent reduction in the size of extracellular calcium nodules deposited by osteoblasts (Fig. 2b, upper panel), as well as by the reduced absorbance of the alizarin red dye extracted from the nodules (Fig. 2b, lower panel and histogram). These effects were not due to differences in cell numbers since ALP activity was corrected by cellular protein content, and the number of cells did not significantly change at the restrictive temperature with EV treatments (untreated cells: $5.14 \times 10^5 \pm 0.13$; SM-EV (tryptase < 110 ng/ml)-treated: $4.92 \times 10^5 \pm 0.16$; SM-EV (tryptase > 110 ng/ml)-treated: $4.85 \times 10^5 \pm 0.11$; mean ± SD) (see also Fig. 2b, upper panel for visualization of cell confluency). Concomitant with the reduction in ALP and calcium minerals, the expression of osteoblast differentiation markers such as RUNX2, the production of OPN and COL1, as well as the phosphorylation of AKT and ERK1/2, which increase during

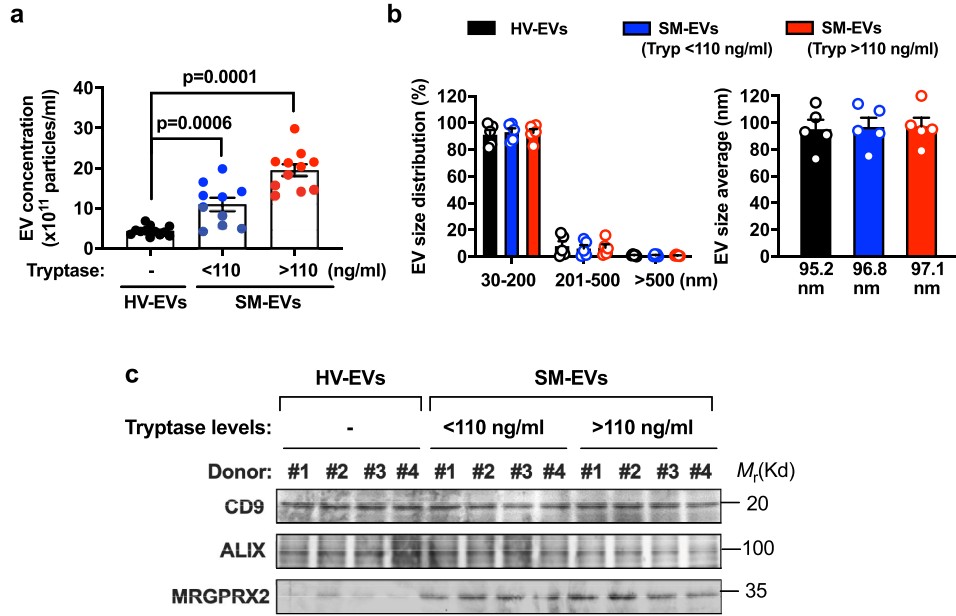

**Fig. 1 Small extracellular vesicles are increased in the serum of patients with systemic mastocytosis. a** Quantitative analysis of extracellular vesicles (EVs) isolated from individual serum samples of healthy volunteers (HV-EVs; black circles; $n = 12$ subjects), patients with systemic mastocytosis (SM) with tryptase values lower than the median (SM-EVs (Tryp < 110 ng/mL); blue circles; $n = 10$) or SM-EVs with tryptase values greater than the median (SM-EVs (Tryp > 110 ng/mL); red circles; $n = 11$) (see Group 1 of Supplementary Table 1 for patient population characteristics). EVs were counted and analyzed using a NanoSight NS300 system. Data represent the mean ± SEM. *P* values using a Mann–Whitney test (two-tailed) are indicated. **b** Size distribution (left) and average size (right) of HV-EVs and SM-EVs as determined by NanoSight. The bars represent five independent measurements ($n = 5$) of pooled HV-EVs from all subjects shown in **a** (black bars), pooled EVs from all patients with SM-EVs (Tryp < 110 ng/ml) (blue bars), and pooled EVs from all patients with (SM-EVs (Tryp > 110 ng/ml); red bars). Data represent the mean ± SEM. Similar results were obtained from at least three separate preparations of EV pools. **c** Western blots illustrating similar expression of EV markers such as CD9 and ALIX in EVs isolated from HV and patients with SM, and enriched expression of mast cell markers such as MRGPX2 in SM-EVs. Equal amounts of protein were loaded in all lanes. Similar results were obtained in at least two additional preparations.

osteoblast differentiation[25], were markedly reduced in cultures treated with SM-EVs (Fig. 2d). These data demonstrate a profound attenuating effect of SM-EVs on the differentiation of hFOB1.19 into bone-producing cells, even after short-term treatment (Fig. 2a).

We then isolated EVs from fresh BM aspirates obtained from three patients with SM, as these EVs might be more representative of EVs in the bone environment. Similar to SM-EVs isolated from serum, EVs derived from BM had the potential to markedly reduce ALP activity and RUNX2 expression when added to osteoblast cultures (Supplementary Fig. 2a), consistent with the conclusion that components of EVs affecting osteoblast differentiation are present both in BM and in circulation.

We also tested the effects of SM-EVs in primary cultures of human mesenchymal stem cells (hMSCs) induced to differentiate into osteoblasts[26] (Fig. 3a). As expected, mesenchymal stem cell (MSC) cultured with osteoblast-inducing media acquired osteoblast-like phenotypes within 9–15 days as determined by the increasing production of ALP that plateaued within 10 days (Fig. 3b, open circles) and by the increased mRNA expression of RUNX2 (Fig. 3c), phenotypes that were not observed when MSC were cultured in normal growth media (Fig. 3b, lower panel and Fig. 3c, white bars). Similar to hFOB1.19 cells, treatment of MSCs with SM-EVs during the course of differentiation in osteoblast-inducing media significantly reduced ALP activity (Fig. 3b, upper panel) and RUNX2 expression (Fig. 3d, black bars), indicating that the attenuating effects of SM-EVs on osteoblastogenesis are not restricted to a cell line as a single model of osteoblast differentiation.

In addition, to exclude effects of potential contaminants brought down with EVs using the ExoQuick precipitation

method we examined the effects of SM-EVs or HV-EVs isolated by density gradient fractionation[27] (Supplementary Fig. 2b). Fractions isolated from SM serum corresponding to small EVs, but not fractions separated at higher densities (non-vesicular), caused significant reductions in RUNX2 expression (Supplementary Fig. 2c). In contrast, the fractions isolated from HV serum had no effect. We further defined that fraction numbers 3–6, containing markers of exosomes and microvesicles such as Annexin 1, CD9, CD63, and ALIX, inhibited osteoblast differentiation by 80% as measured by ALP activity, suggesting that the active components of SM-EVs are present primarily in low density EVs (Supplementary Fig. 2d).

**SM-EVs contain bone modifying miRNAs that are taken up by osteoblasts.** We then considered whether miRNAs may contribute to the regulation of osteoblast differentiation by SM-EVs since miRNAs have been implicated in the regulation of osteoblastogenesis and osteoclastogenesis[18,19,28–30] and characteristic miRNA profiles in blood/serum are associated with patients with osteoporotic fractures[31]. From various candidates reported to be linked to bone homeostasis[29,31], qRT-PCR analysis of SM-EVs indicated that miR-21, miR-23-a, miR-25, miR-93, and miR-30a were upregulated in SM-EVs compared to HV-EVs, with fold changes ranging from 1.5 (miR-21) to 40-fold (miR-25) (Fig. 4a). Furthermore, exposure of hFOB1.19 to SM-EVs from either group of patients (with < or >110-ng/ml tryptase values) resulted in significant increases in the levels of all these miRNAs in hFOB1.19 as early as 6–12 h (Supplementary Fig. 3a) and for at least 3 days (Fig. 4b), suggesting that the EVs not only carry these miRNAs, they transfer their miRNA contents into osteoblasts.

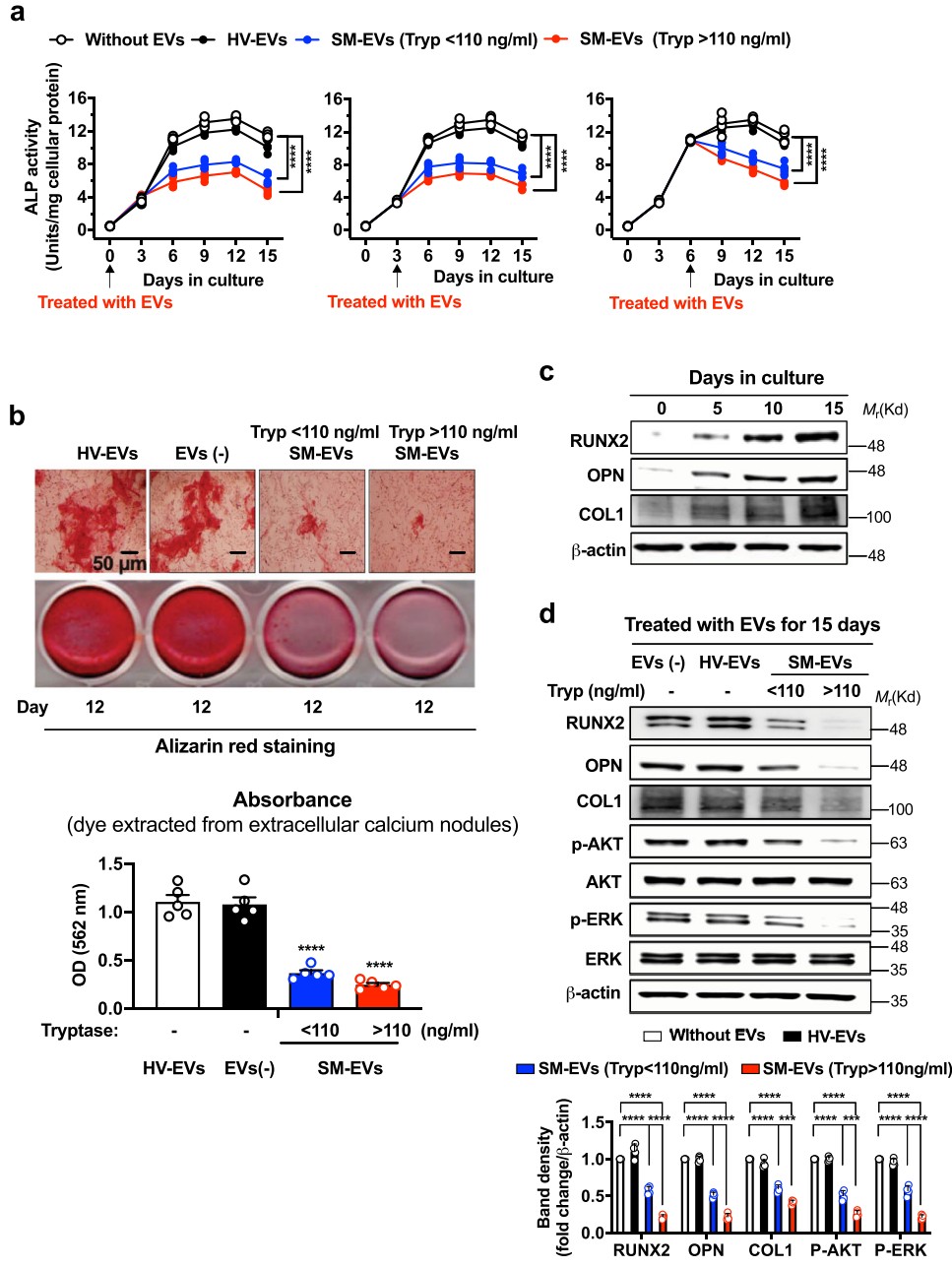

**Fig. 2 SM-derived EVs inhibit differentiation and mineralization of hFOB1.19 pre-osteoblasts. a** Effect of extracellular vesicles (EVs) isolated from the serum of healthy volunteers (HV-EVs) or of patients with systemic mastocytois (SM-EVs) on the differentiation of hFOB1.19 osteoblasts. Osteoblasts were plated at 39.5 °C and after 2 h (left panel), 3 days (middle panel) or 6 days (right panel), treated with a pool of EVs from the indicated groups (100 μg). Media were replaced every 3 days and EVs replenished at the same time. Supernatants were collected to measure alkaline phosphatase (ALP) activity. Results are expressed as units per mg of cellular protein in each well. A two-way ANOVA was used for statistic comparison of the curves. ****$p < 0.0001$ compared to untreated cultures. Individual data points are shown and often overlap ($n = 5$ biological replicates in a representative experiment). Empty circles are untreated cells; black circles, cells treated HV-EVs; blue and red circles are, respectively, cells treated with SM-EVs of patients with serum tryptase lower (SM-EVs (Tryp < 110 ng/ml)) or greater (SM-EVs (Tryp > 110 ng/ml)) than the median (110 ng/mL). **b** Effect of SM-EVs as defined above on calcium deposition by osteoblasts using alizarin red staining. hFOB1.19 were plated and treated with EVs as in **a** (left panel) for 12 days. The top picture shows osteoblasts under a microscope (20×) to illustrate the effect of EVs on extracellular calcium deposition (visualized in red). Scale bar is 50 μm. The picture underneath is a macroscopic picture of the wells during extraction of the dye from the calcium nodules. The extracted dye was quantified by absorbance at 562 nm (bar graphs). Data show the mean ± SEM of $n = 5$ biological replicates of a representative experiment. ****$p < 0.0001$ using an unpaired Student $t$-test (two-tailed). **c** Changes in RUNX2, OPN, and COL1 expression during differentiation of hFOB1.19 determined by Western blotting. **d** Effect of HV-EVs and SM-EVs (as defined in **a** and using the same color scheme, also indicated in the legend), on the expression of RUNX2, OPN, and COL1 and phosphorylation of AKT and ERK. The histograms represent band intensity changes shown as mean ± SD of $n = 4$ independent experiments. In **c** and **d**, β-actin was used as a loading control. Total AKT and ERK are also shown in **d**. ***$p < 0.001$ ($p = 0.0008$ in COL1 and $p = 0.0009$ in p-AKT) and ****$p < 0.0001$, using an unpaired Student $t$-test (two-tailed). Data in **a**–**d** are from representative experiments of at least three independent experiments showing similar effects.

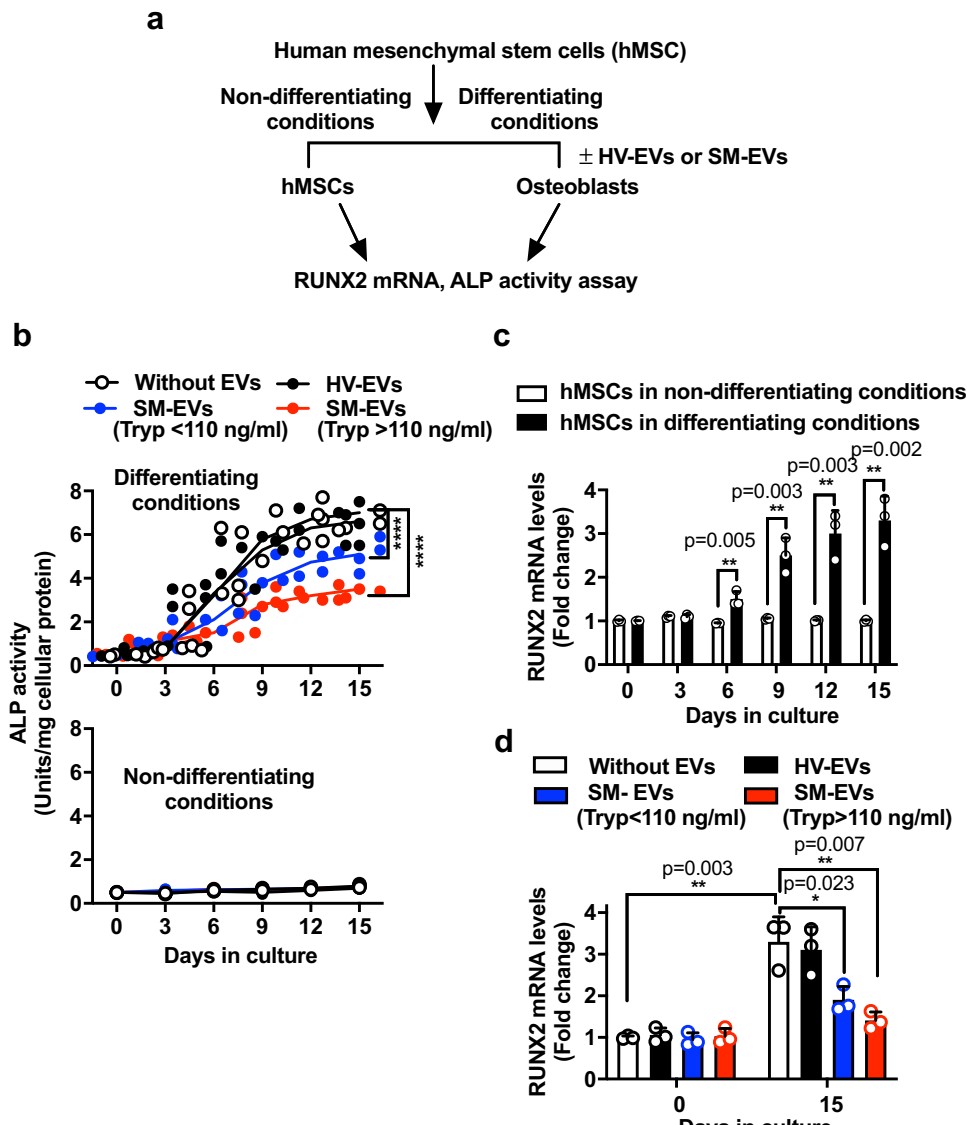

**Fig. 3 SM-derived EVs inhibit differentiation of human mesenchymal stem cells into osteoblasts. a** Protocol for differentiation of human mesenchymal stem cells (hMSC) into osteoblasts. **b** Effect of extracellular vesicles (EVs) isolated from the serum of healthy volunteers (HV-EVs) or of patients with systemic mastocytois (SM-EVs) on the differentiation of hMSCs into osteoblasts when cultured in osteoblast differentiation media for 15 days (upper panel) or in normal growth media (lower panel). EVs (100 μg) were added to the cultures every 3 days. Empty circles are untreated cells; black circles, cells treated with a pool of HV-EVs; blue and red circles are, cells treated with a pool of SM-EVs of patients with serum tryptase values lower or greater than 110 ng/mL (SM-EVs (Tryp < 110 ng/ml) and SM-EVs (Tryp > 110 ng/ml), respectively). Supernatants were collected to measure alkaline phosphatase (ALP) activity and the activity corrected by the amount of cellular protein in the wells. A two-way ANOVA was used for statistic comparison of the curves. ****$p < 0.0001$ compared to untreated cultures. Individual points are shown ($n = 5$ biological replicates). **c** Changes in RUNX2 cellular mRNA expression overtime in osteoblast differentiation media (black bars) or normal growth media (white bars) ($n = 3$). **d** Effect of HV-EVs or SM-EVs as defined above on the mRNA expression of RUNX2 in hMSC-derived osteoblasts 15 days after plating in osteoblast differentiation media ($n = 3$). Cells were treated with EVs as in **b**, and the levels of RUNX2 mRNA quantified using qRT-PCR and normalized to GAPDH expression levels. The color scheme is as described in **b** and indicated in the figure. Data in **c** and **d** are expressed as fold change compared to untreated cells right after plating at day 0 and show the mean ± SD of a representative experiment. Unpaired Student $t$-tests (two-tailed) were used for the indicated statistical comparisons. *$p < 0.05$; **$p < 0.01$. In **b**–**d**, $n$ indicates biological replicates (independent wells) in a representative experiment. Similar results were repeated in two additional experiments.

To gain further insights into their potential role, we then examined the correlations between the levels of these miRNAs in individual EV preparations from patients and the inhibition they exerted in osteoblasts differentiation. As shown in Fig. 4c, in a total of 31 patients with characteristics shown in Supplementary Table 1, Groups 1 and 2, the levels of miR-21, miR-23a, and miR-30a contained in SM-EVs but not those of miR-25 and miR-93, significantly correlated with their inhibitory effect on ALP activity, suggesting a potential role, particularly for miR-23a

and miR-30a, in osteoblast differentiation. Consistent with this conclusion, the levels of miR-23a and miR-30a were significantly higher in EVs from patients with osteopenia/osteoporosis (Supplementary Fig. 3b) in the patient cohorts in Supplementary Table 1 (Groups 1 and 2). The higher levels of miR-23a and miR-30a in the osteoporosis positive group were not related to gender as these levels were similar between males and females (Supplementary Fig. 3c). Furthermore, the levels of miR-23a and miR-30a in EVs from three patients in our cohort with

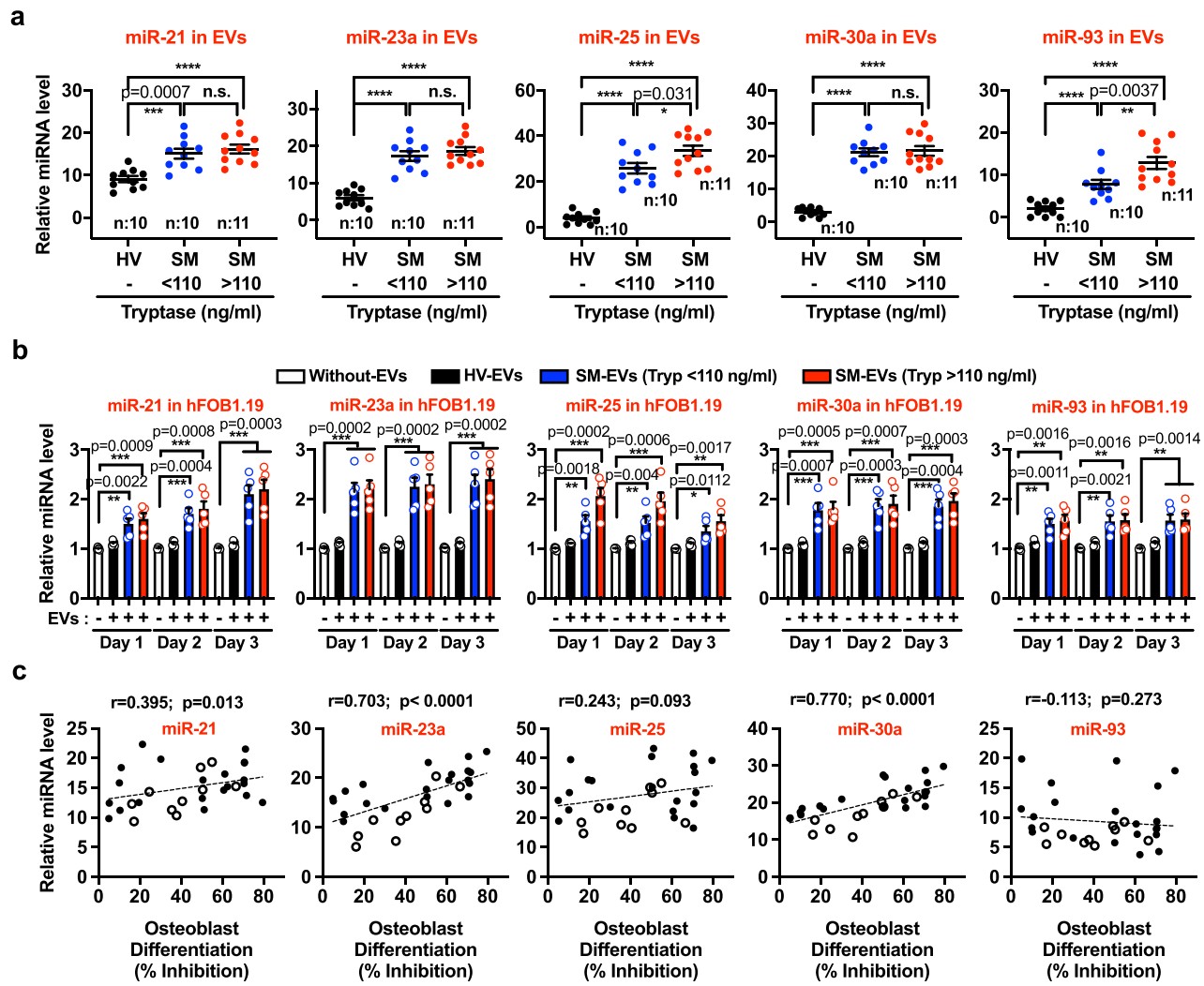

**Fig. 4 miRNAs involved in osteoblast fate regulation are elevated in SM-EVs and transferred to osteoblasts. a** Levels of miR-21, miR-23a, miR-25, miR-30a, and miR-93 in extracellular vesicles (EVs) isolated from the serum of healthy volunteers (HV) or of patients with systemic mastocytois (SM) with serum tryptase values lower or greater than 110 ng/mL (SM < 110 ng/ml and SM > 110 ng/ml, respectively), as indicated. RNA was extracted from 100 μg of EVs isolated from each patient. The expression levels of the indicated miRNA species were measured by qRT-PCR and normalized to the levels of a spike-in control (miR-39). Individual data points are shown. Exact p values using a Mann–Whitney test (two-tailed) are indicated and **** represents p < 0.0001. **b** Levels of miR-21, mR-23a, miR-25, miR-30a, and miR-93 in osteoblasts after treatment with EVs isolated from the serum of healthy volunteers (HV-EVs) or of patients with SM (SM-EVs). hFOB1.19 were plated and treated once with 100 μg of EVs pooled from all subjects from each group for 1, 2, or 3 days at the restrictive temperature (39.5 °C). Cells were then washed and equal amounts of mRNA reversed-transcribed. MiRNA levels were determined by qRT-PCR and normalized to the levels of GAPDH. As indicated in the figure, empty bars are untreated cells; black bars, cells treated with HV-EVs; blue and red bars, cells treated, respectively, with SM-EVs of patients with serum tryptase values lower (SM-EVs (Tryp < 110 ng/ml)) or >110 ng/mL (SM-EVs (Tryp > 110 ng/ml)). Data are expressed as fold change in comparison to untreated cells each day, and shown as the mean ± SEM (n = 5 biological replicates of a representative experiment). Similar results were obtained in two additional experiments. *p < 0.05; **p < 0.01; and ***p < 0.001, using unpaired Student t-tests (two-tailed). **c** Pearson's correlations between the relative miRNA levels of miR-21, miR-23a, miR-25, miR-30a, and miR-93 in individual patient's EVs and their respective effect on the inhibition of osteoblast differentiation after 12 days of treatment. These include patients shown in **a** and summarized in Group 1 of Supplementary Table 1 (closed circles); and ten additional patients with characteristics and demographics shown in Group 2 of Supplementary Table 1 (open circles).

osteopathy and smoldering SM, a more aggressive form of SM, were significantly increased compared to those of patients with indolent disease and osteopathy (Supplementary Fig. 3d) and all three caused >70% inhibition of osteoblast differentiation in culture, suggesting that this mechanism may also be at play in more aggressive variants.

**miR-23a and miR-30a in SM-EVs inhibit osteoblast differentiation**. We transfected hFOB1.19 with mimic sequences for miR-21, miR-23-a, miR-25, miR-30a, miR-93, or a control

sequence (with no known target) to further investigate their effects in osteoblastogenesis. Transfection of osteoblasts with these miRNA mimics increased the levels of the corresponding miRNAs in osteoblasts to a level comparable to that seen after treatment with SM-EVs (compare Supplementary Fig. 4 with Fig. 4b). Transfection with miR-23a or miR-30a mimics caused inhibition in ALP activity (Fig. 5a and Supplementary Fig. 5b, d) concomitant with a reduction in RUNX2 expression (Fig. 5b), as found after treatment with SM-EVs (Figs. 2d and 3d). However, miR-25 and miR-93, which has been described as inhibitor of

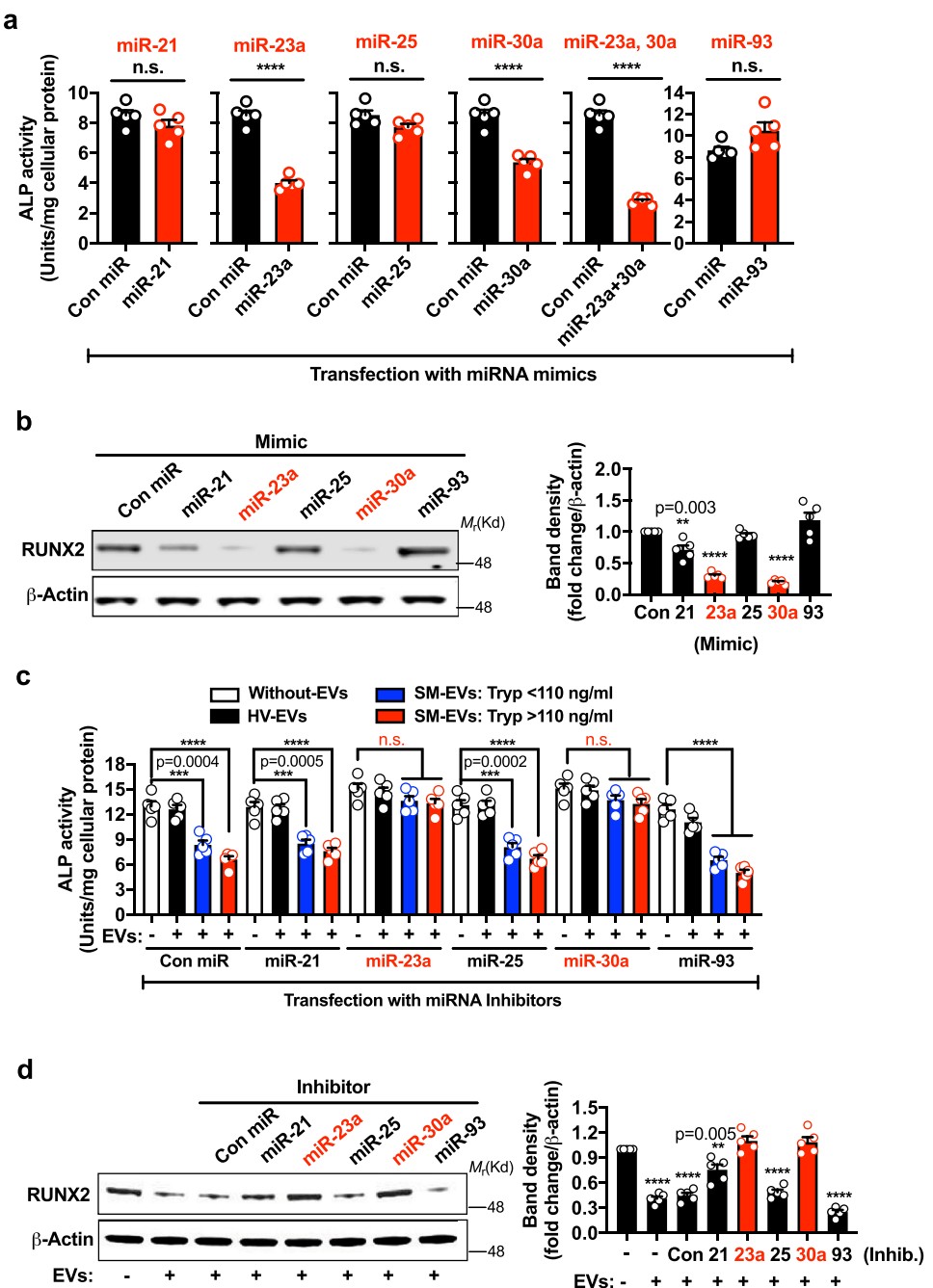

**Fig. 5 miR-23a and miR-30a mediate inhibition of osteoblast differentiation induced by SM-EVs. a, b** Effect of miRNA mimics on osteoblast differentiation. hFOB1.19 cells were plated and next day transfected with 100 nM of control (Con miR), miR-21, m-R23a, miR-25, miR-30a, or miR-93 mimics. After 2 days, the transfection media were replaced. Five days later, alkaline phosphatase (ALP) activity was determined in the culture media (**a**) and RUNX2 protein expression in cell lysates determined by Western blot (**b**). **c, d** Functional knockdown of miR-23a and miR-30a reverse the attenuating effects of extracellular vesicles isolated from the serum of patients with systemic mastocytois (SM-EVs) on osteoblast differentiation. hFOB1.19 cells were seeded and then treated or not (white bars are untreated cells) with pooled extracellular vesicles isolated from the serum of healthy volunteers (HV-EVs) (black bars), pooled SM-EVs from patients with serum tryptase values lower than 110 ng/mL (blue bars; SM-EV: Tryp < 110 ng/mL), or SM-EVs from patients with serum tryptase values >110 ng/mL (red bars; SM-EV: Tryp > 110 ng/mL), as described in "Methods." Cells were then transfected with the miRNA inhibitors (inhib.) for 2 days. Transfection media were replaced and 5 days later, ALP activity was measured in the media (**c**) and RUNX2 expression in cells detected by Western blotting (**d**). In **b** and **d**, the bar graphs represent the relative fold changes in band intensity normalized to β-actin and compared to the respective controls (first bar). All data (**a**–**d**) represent the mean ± SEM ($n = 5$ biological replicates) of representative experiments and repeated at least twice with similar results. In **a**–**d**, **$p < 0.01$; ***$p < 0.001$; ****$p < 0.0001$; and n.s., not significant using an unpaired Student $t$-test (two-tailed) for the indicated comparisons or compared to the first bar (**b**, **d**). The blots are representative images from one experiment.

osteoblast differentiation[32], did not alter ALP activity or RUNX2 expression (Fig. 5a, b and Supplementary Fig. 5c, e). In addition, the miR-21 mimic reduced RUNX2 expression, but less prominently than miR-23a and miR-30a (Fig. 5b), and only caused a temporary and minor reduction in ALP activity 1 day after transfection (Supplementary Fig. 5a) but not 5 days after transfection (Fig. 5a).

Transfections with combinations of the five miRNA mimics and consequent increases in their respective miRNA levels in osteoblasts (Supplementary Fig. 6a), inhibited ALP activity to a similar extent as SM-EVs (compare Fig. 2 with Supplementary Fig. 6b). While miR-23a and miR-30a in combination caused stronger inhibition of osteoblast differentiation than each individually (Fig. 5a), miR-93 together with miR-23a and/or miR-30a, lessened their inhibitory effects (Supplementary Fig. 6b). The results suggest that SM-EVs contain miRNA species with the ability to regulate osteoblastogenesis and identify miR-23a and miR-30a as strong candidates restricting osteoblast differentiation/maturation.

To demonstrate that these miRNAs mediate the attenuating effect of SM-EVs on osteoblast differentiation, we silenced miR-21, miR-23-a, miR-25, miR-30a, or miR-93 in osteoblasts after treatment with SM-EVs. Transfection of hFOB1.19 cells with chemically modified single-stranded antisense RNA molecules (miR inhibitors) reduced the levels of corresponding miRNAs after SM-EV treatment (Supplementary Fig. 7a–e). Inhibition of either miR-23a or miR-30a effectively reversed the reduction in osteoblast ALP activity (Fig. 5c) and RUNX2 expression (Fig. 5d) induced by SM-EVs. However, knockdown of the other miRNAs did not show significant effects (Fig. 5c, d). The nullifying effects of miR-23a and miR-30a knockdown on SM-EV-mediated effects are specular images of the effects of miR-23a and miR-30a mimics, strongly implicating miR-23a and miR-30a as major culprit mediators of the effects of SM-EVs on osteoblast differentiation/maturation.

**HMC-EVs inhibit osteoblast differentiation via miR-23a and miR-30a.** We also determined the levels of miR-21, miR-23-a, miR-25, miR-30a, and miR-93 in EVs isolated from the culture media of neoplastic mast cell lines (HMC-1.1 and HMC-1.2), since SM-EVs contain mast cell markers that suggest a mast cell origin (Fig. 1c and ref. [22]). In contrast to the transformed epithelial cell line HEK293, which sheds abundant EVs, EVs from HMC-1.1 (expressing the V560G-KIT variant) and particularly HMC-1.2 (expressing V560G-KIT and D816V-KIT, the most common variant associated with SM), produced substantial amounts of miR-21, miR-23-a, miR-25, miR-30a, and miR-93 (Fig. 6a). Similar to SM-EVs, treatment of hFOB1.19 osteoblasts with EVs derived from HMC-1.1 and HMC-1.2 mast cells induced a concentration-dependent inhibition of hFOB1.19 differentiation (Fig. 6b, c, upper panels) and mineralization (Fig. 6b, c, lower panels).

Since HMC-1.2-EVs contained greater levels of miR-23-a and miR-30a than HMC-1.1-derived EVs (Fig. 6a) and caused stronger inhibition of osteoblast differentiation and mineralization (Fig. 6b, c), we transfected HMC-1.2 cells with miRNA inhibitors for miR-23a and miR-30a or control oligos to determine their role on osteoblast maturation. EVs shed from miR-23a and miR-30a-knockdown HMC-1.2 cells showed 50% lower concentrations in these miRNAs (Fig. 6d) and, concomitantly, they were 50% less effective in inhibiting osteoblast differentiation (Fig. 6e). The data demonstrate that EVs derived from neoplastic mast cells, as those derived from patients with SM, contain osteoblast fate-modifying miRNAs, namely miRNA-23a and miRNA-30a, with the ability to hinder osteoblast differentiation.

**SM-EVs and miRNA-23a and miRNA-30a reduce RUNX2 and SMAD1/5 expression.** Based on previous studies on targets for miR-23a and miR-30 in osteoblasts[33–35] and computational predictions of mRNA targets (obtained using miRSystem[36] http://mirsystem.cgm.ntu.edu.tw), we tested the effects of SM-EVs on the expression of several potential genes critical for osteoblast differentiation. Treatment of FOB1.19 with SM-EVs, significantly reduced mRNA levels of RUNX2, SMAD1, SMAD5, and to a lesser extent SMAD2, but did not affect special AT-rich sequence-binding protein 2 (SATB2) or C-X-C Motif Chemokine Ligand 12 (CXCL12) expression (Fig.7a). The reductions in RUNX2 and SMADs mRNA levels were associated with concomitant reductions in protein expression (Figs. 2c and 7b). Furthermore, the expression of BMP2, known to activate SMADs and to regulate and to be regulated by RUNX2 in a positive feedback regulatory loop, was also markedly diminished by treatment with SM-EVs at the mRNA (Fig.7a) and protein levels (Fig. 7b).

We then determined whether the expression of RUNX2 and SMAD1/2/5 could also be affected by miR-23a or miR-30a mimics. As shown in Fig. 7c, both miR-23a and miR-30a similarly reduced the expression of RUNX2, although they differed in their relative specificity for SMAD proteins. While miR-23a caused a marked inhibition in SMAD5 mRNA levels and a slight inhibition in SMAD1 with no effect in SMAD2, miR-30a inhibited the expression of SMAD1 more than SMAD5 with a modest effect on SMAD2. The inhibition of RUNX2 and SMAD5 expression was maximal 1–5 days after transfection, while maximal inhibition of SMAD 1 and 2 was generally more pronounced 5 days after transfection (Fig. 7c). The relative specificity of miR-23a and miR-30a for SMAD1, 2, and 5 suggests that the actions on SMADs are not merely secondary to RUNX2, as both miRNAs inhibited RUNX2 similarly. These data, together with previous reports and algorithmic predictions point toward RUNX2 and SMAD proteins as target genes for SM-EVs at least partly via transfer of miR-23a and miR-30a. As the effects of SM-EVs from the high tryptase group were more pronounced than those from the lower tryptase group (Figs. 2d, and 7a, b) even though the levels of miR-23a and miR-30a were similar (Fig. 4a), other EV components, related or unrelated to the different gender distribution between the groups, may also contribute to these effects.

**BM from patients with SM and bone disease have lower RUNX2.** To better understand the implications of our findings in our patient population, we compared the effects of EVs derived from patients with or without bone disease on osteoblast differentiation. As shown in Fig. 8a, SM-EVs from patients with osteopenia/osteoporosis (patient characteristics shown in Groups 1 and 2 of Supplementary Table 1) caused significantly greater inhibition of osteoblast differentiation in culture than SM-EVs from patients with no osteopenia/osteoporosis, and contained higher levels of miR-23a and miR-30a (Supplementary Fig. 3b), potentially linking such effects to bone disease in SM.

We also extracted mRNA from cells in BM aspirates of patients with or without osteopenia/osteoporosis (described in Group 3 of Supplementary Table 1) and found that the expression levels of RUNX2 were reduced in those with osteopenia/osteoporosis (Fig. 8b), consistent with reduced osteoblast differentiation in association with bone disease. No differences in ALP mRNA levels were however observed between these subpopulations, suggesting that in these BM cells, ALP mRNA expression may also be regulated by RUNX2-independent mechanisms.

In regards to the expression of bone remodeling and osteoclastogenesis markers, RANKL mRNA levels were reduced while OPG was slightly increased, and thus the ratio RANKL/OPG was markedly

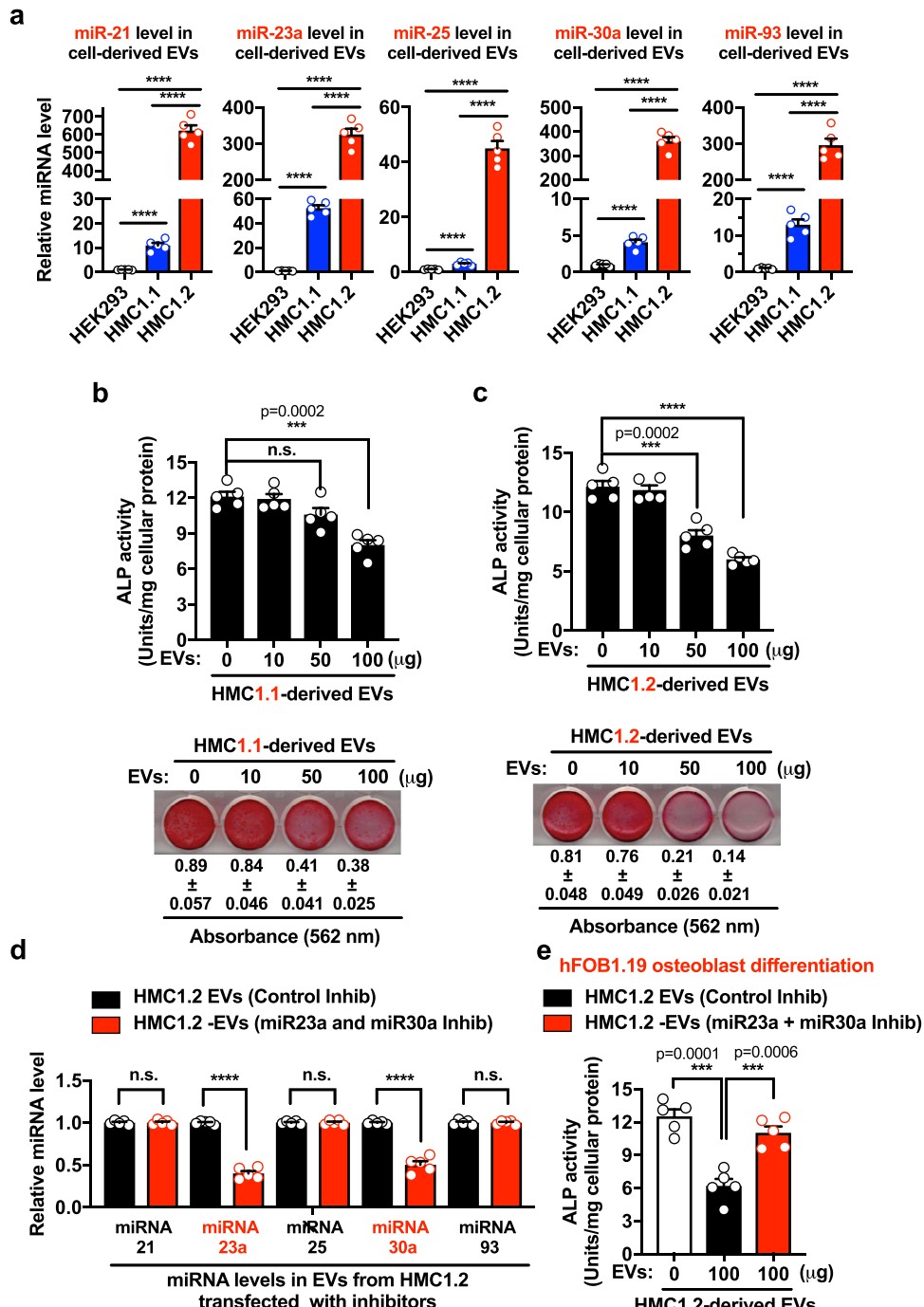

**Fig. 6 Neoplastic mast cells-shed EVs inhibit osteoblast differentiation via miR-23a and miR-30a. a** Expression of miR-21, mR-23a, miR-25, miR-30a, and miR-93 in extracellular vesicles (EVs) released from HEK293, or the neoplastic mast cell lines HMC-1.1 and HMC-1.2, as indicated. RNA was extracted from 100 μg of EVs isolated from the supernatants of the cell cultures and expression levels of the indicated miRNA species (from equal amounts of total RNA) were measured by qRT-PCR and expressed as fold change of the content in EVs from HEK293 cells. **b, c** Effect of EVs derived from HMC-1.1 (**b**) or HMC-1.2 (**c**) in the differentiation and mineralization of osteoblasts. hFOB1.19 were plated and treated with 10, 50, or 100 μg of EVs. At the end of 12 days, alkaline phosphatase (ALP) activity (bar graphs in **b** and **c**) was measured. The presence of calcium deposits by staining with alizarin red staining was also determined (**b** and **c**, lower panels). **d** Levels of miRNAs in the EVs of HMC-1.2 cells after transfection of these cells with antisense oliogonucleotides inhibitors (inhib.) for miR-23a and miR-30a (miR-23a and miR-30a inhib; red bars) or with control oligos (control inhib; black bars). Levels of the indicated miRNAs were determined by qRT-PCR as in **a**. **e** Effect of EVs derived from miR-23a- and miR-30a-silenced HMC-1.2 cells on osteoblast differentiation. Osteoblast (hFOB1.19) cultures were treated or not (white bar, untreated cells) with 100 μg of EVs from HMC-1.2 transfected with control oligos (HMC-1.2-EVs (Control inhib.), black bar) or HMC-1.2 transfected with miR-23a and miR-30a inhibitors ((HMC-1.2-EVs (miR-23a + miR-30a inhib.), red bar). ALP activity was determined after 6 days. Data in **a–e** are the mean ± SEM ($n = 5$ biological replicates) of a representative experiment out of three experiments with similar results. In **a–e** ***$p < 0.001$; ****$p < 0.0001$; and n.s., not significant, using unpaired Student $t$-tests (two-tailed).

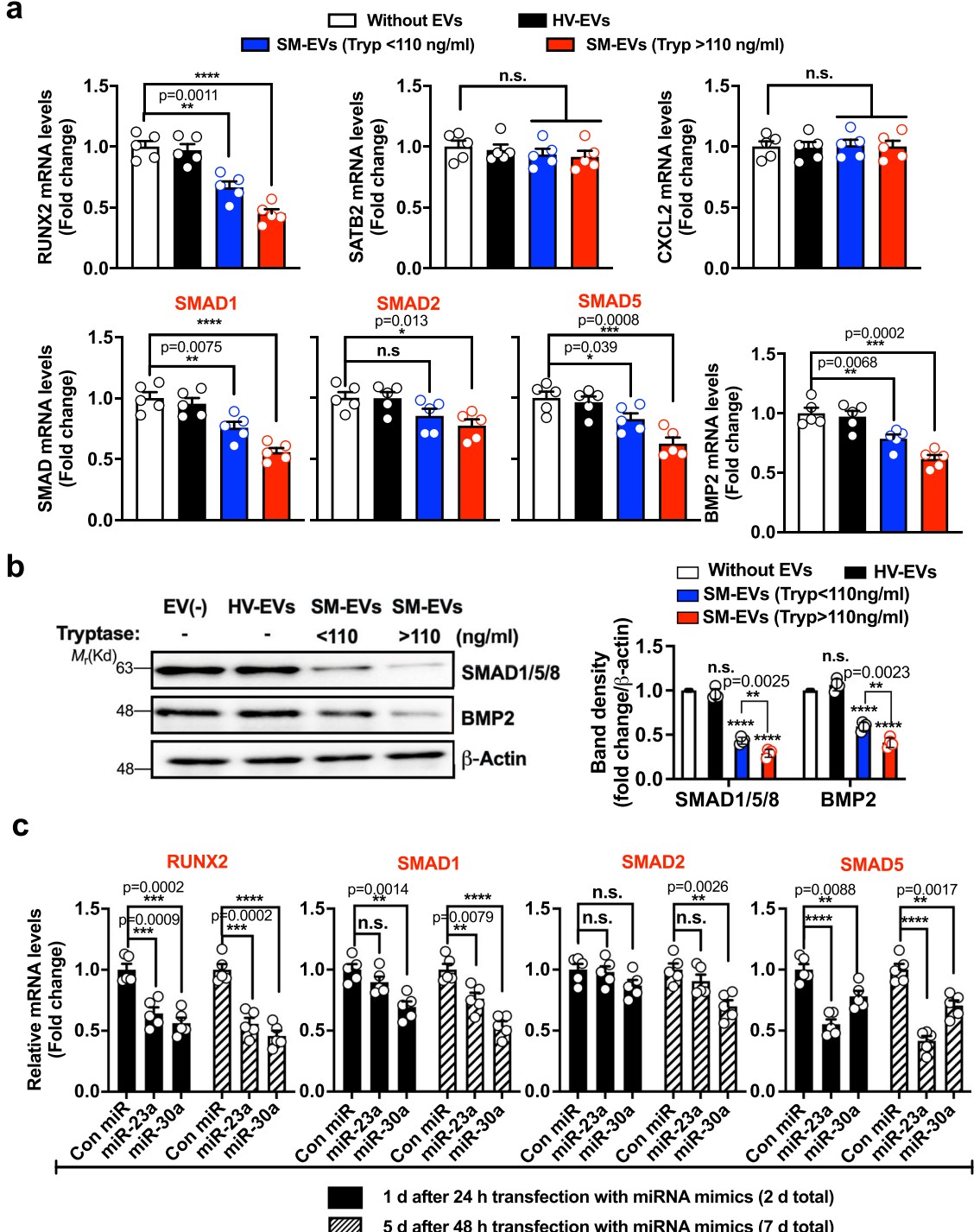

**Fig. 7 SM-EVs as well as miR-23a and miR-30a mimics target RUNX2 and SMAD1/5. a** Changes in mRNA levels of RUNX2, SATB2, CXCL12, SMAD1/2/5, and BMP2 in hFOB1.19 cells after treatment for 12 days with 100 µg of pooled extracellular vesicles (EVs) from healthy volunteers (HV-EVs; black bars), patients with systemic mastocytosis (SM-EVs) with tryptase <110 ng/ml (blue bars), or patients with tryptase >110 ng/ml (red bars) or left untreated (white bars). The expression levels of the indicated mRNA were measured by qRT-PCR. Expression of GAPDH was used for normalization and the results are shown as fold change compared to untreated cultures. **b** Effect of SM-EVs on the expression of SMADs and BMP2 during differentiation of hFOB1.19 cells. β-Actin was used as a loading control. Histograms represent band intensity changes, shown as mean ± SD of $n = 4$ independent experiments, following the same color scheme as in **a** and indicated in the figure. Uncropped blots are provided in the Source Data file. **c** Changes in the mRNA levels of RUNX2, and SMAD1/2/5 after transfection with mimic RNA for miR-23a, miR-30a, or control mimic (Con miR) in hFOB1.19 cells. 24 or 48 h after transfection, the media were replaced, and cellular RNA was extracted 1 day (black bars) or 5 days later (hatched bars), as indicated. Expression levels of the indicated transcripts were determined as in **a**. Data in **a** and **c** represent the mean ± SEM ($n = 5$ biological replicates) of a representative experiment out of three independent experiments with similar results. In all panels: *$p < 0.05$; **$p < 0.01$; ***$p < 0.001$; ****$p < 0.0001$; n.s., not significant, using an unpaired Student $t$-test (two-tailed) for the indicated comparisons.

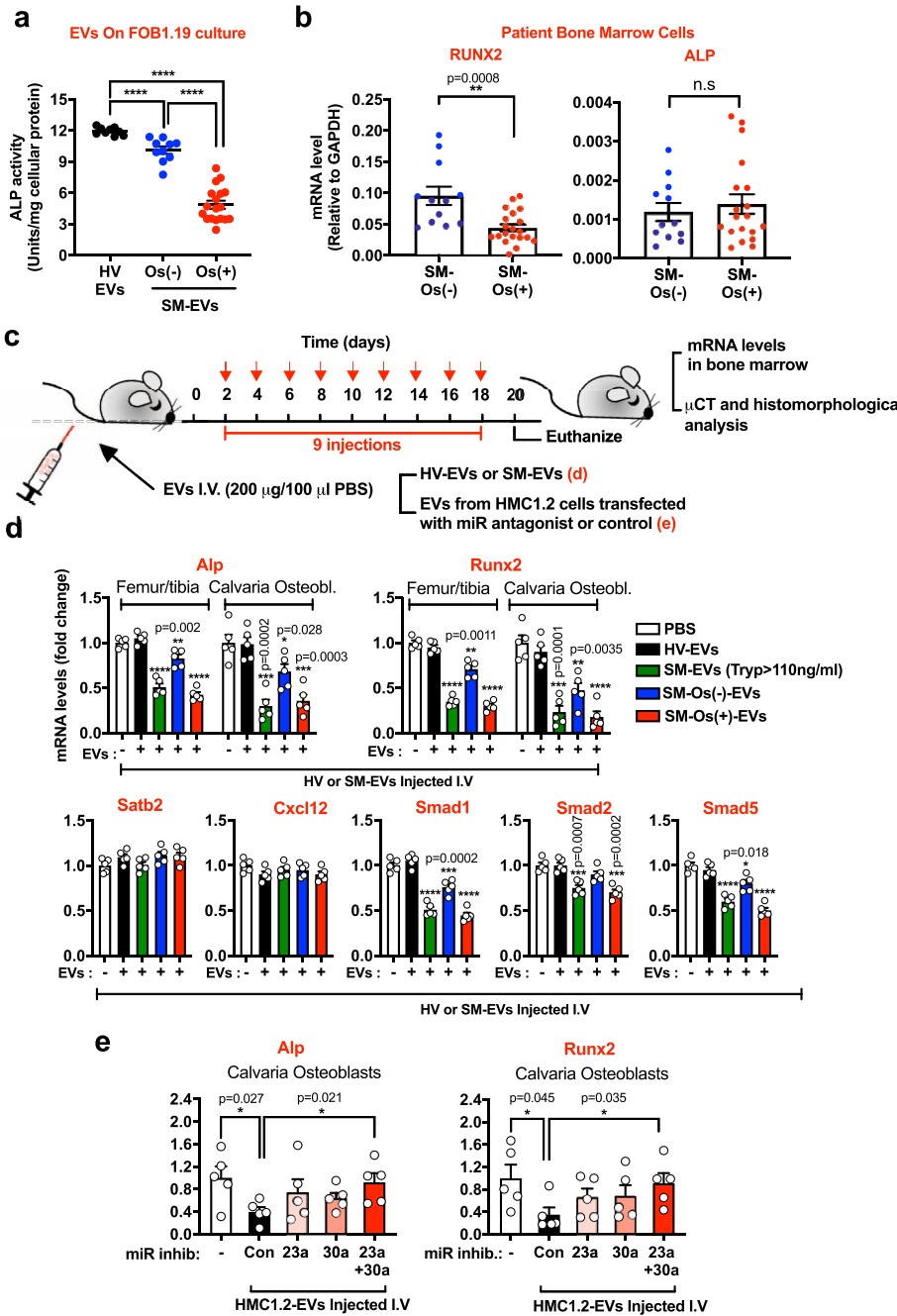

diminished in patients with SM-osteopenia/osteoporosis compared to patients without osteopenia/osteoporosis (Supplementary Fig. 8a). The reduced RANKL/OPG mRNA ratio does not thus support increased osteoclastogenesis in these patients, although it is important to note that the RANKL/OPG mRNA ratio in cells from BM aspirates may not necessarily reflect their protein ratios or represent the collective BM regulatory microenvironment. In addition, the expression levels of TRAP (ACP5) and RANK, expressed by osteoclasts, were unchanged or reduced, respectively, in these patients compared to those without osteopenia/osteoporosis (Supplementary Fig. 8b). Even though the mRNA data in BM from patients suggest a complex regulatory overall environment with altered bone remodeling, the reduced levels of RUNX2 and RANKL mRNAs agree with the conclusion that in SM with bone disease, osteoblast differentiation is attenuated.

**SM- or HMC-1.2-EVs reduce bone formation markers in mice.** Next, we investigated the potential long-term effects of SM-EVs on osteogenesis in vivo by injection into recipient mice for 2–3 weeks (Fig. 8c). BM cells from the femurs and tibias of recipient mice injected with SM-EVs (from patients with tryptase >110 ng/ml) or SM-EVs from patients with osteopenia/osteoporosis showed markedly reduced mRNA expression levels of Alp, Runx2, Smad1/5, and to a lesser extent Smad2 (Fig. 8d). Similar reductions in Alp and Runx2 mRNA expression were also found in osteoblasts isolated from the calvaria of these mice (Fig. 8d). However, expression of these transcripts was less affected by SM-EVs pooled from patients with no bone symptoms and unaffected in mice injected with a pool of EVs from HV (Fig. 8d). The expression of Satb2 and Cxcl12 did not change in any of the injected groups of mice (Fig. 8d), in agreement with the

**Fig. 8 Osteoblast differentiation markers are reduced in BM from patients with SM-osteopenia/osteoporosis, and in mice treated with SM-EVs from these patients. a** Effect of extracellular vesicles (EVs) isolated from the serum of healthy volunteers (HV-EVs; black circles; $n = 9$ subjects) or of patients with systemic mastocytois (SM-EVs) diagnosed with (Os(+); red circles; $n = 18$) or without osteopenia/osteoporosis (Os(−); blue circles; $n = 10$) (from Groups 1 and 2 of Supplementary Table 1) on alkaline phosphatase (ALP) activity as a measure of osteoblast differentiation. hFOB1.19 were treated with the indicated EVs from individual subjects (100 µg) 2 h after plating and then every 3 days for 12 days. ALP activity in the culture supernatants was measured and corrected by protein in the cell lysates. Data represent the mean ± SEM. ***$p < 0.001$; and ****$p < 0.0001$ using an unpaired Student $t$-test (two-tailed). **b** Expression levels of ALP and RUNX2 in cells from bone marrow (BM) aspirates of patients with osteopenia/osteoporosis (SM-Os(+); red circles; $n = 19$) or without osteopenia/osteoporosis (SM-Os(−); blue circles; $n = 12$) measured by qRT-PCR and normalized by GAPDH expression levels. Details on the patients are in Group 3 of Supplementary Table 1. Data represent the mean ± SEM. **$p < 0.01$; and n.s., not significant using a Mann–Whitney test (two-tailed). **c** Scheme illustrating the regimen of injection of HV-EVs or SM-EVs into recipient mice. **d**, **e** Expression of transcripts for Alp, Runx2, Satb2, Cxcl12, and Smad1/5, as indicated, in BM cells flushed from femurs/tibias or extracted from calvaria's osteoblasts of mice ($n = 5$) injected with PBS (white bars), HV-EVs (black bars), SM-EVs pooled from patients with serum tryptase values >110 ng/mL (SM-EVs (Tryp >100 ng/ml; green bars), SM-EVs pooled from patients without osteopenia/osteoporosis (SM-Os(−)-EVs; blue bars), or SM-EVs pooled from patients with osteopenia/osteoporosis (SM-Os(+)-EVs; red bars) (**d**); or EVs released by HMC-1.2 cells into the culture media (HMC-1.2-EVs) (**e**), as indicated in **c**. In **e**, HMC-1.2-EVs were isolated from the culture media of HMC-1.2 cells transfected or not (control; Con) with inhibitors (miR inhib.) for miR-23a, miR-30a, or both, as indicated. Similar results were obtained in a separate experiment. Expression of GAPDH was used for normalization and the results are shown as fold change compared to mice injected with HV-EVs (**d**) or with mice injected with PBS (**e**). Data in **d** and **e** represent the mean ± SEM ($n = 5$). *$p < 0.05$; **$p < 0.01$; ***$p < 0.001$ using an unpaired Student $t$-test (two-tailed).

lack of effects on these transcripts in osteoblasts cultures by SM-EVs and miRNA mimics (Fig. 7a).

Furthermore, mice treated with EVs from neoplastic HMC-1.2 mast cells (as illustrated in Fig. 8c), similarly to those injected with SM-EVs (Fig. 8d), showed reduced expression of Alp and Runx2 in calvaria osteoblasts (Fig. 8e). In contrast, EVs that were isolated from HMC-1.2 after transfection with antagonists for miR-23a, miR-30a, or both, which had reduced levels of these miRNAs (Fig. 6d), were less effective in reducing Alp and Runx2 expression in calvaria osteoblasts, particularly when both miR-23a and miR-30a were knocked down (Fig. 8e). These results demonstrate that SM-EVs as well as HMC-1.2-derived EVs reduce osteoblast markers in vitro and in vivo and that miR-23a and miR-30a are crucial players mediating these effects.

**SM-EVs reduce trabecular bone density when injected into mice.** To gain insight into potential changes in bone morphometry, we also analyzed the long bones of mice after injection of HV or SM-EVs by micro-computed tomography (µCT) (see scheme in Fig. 8c). Quantitative µCT analysis (Fig. 9a, b) showed that injection of SM-EVs from patients with tryptase >110 ng/ml or SM-EVs from patients with osteoporosis/osteopenia induced a 28% decrease in the percent of trabecular bone volume (Tb. BV/TV) with a decrease in the number of trabeculae (Tb.N) and an increase in trabecular separation (Tb. Sp), and concomitant reduction in bone mass density (BMD). In agreement with the µCT analysis, microscopic examination of decalcified proximal tibial bones revealed thicker bone trabeculae in mice injected with HV-EVs compared to those injected with SM-EV and histomorphological analysis indicated a reduced percent of total bone area in mice treated with SM-EVs in the proximal tibial bones (Fig. 9c, d). No other obvious morphological or histopathological differences between treatments were observed (Fig. 9c). Consistent with the reduced expression of Runx2 mRNA in the marrows of mice injected with SM-EVs (Fig. 8d), the percentage of Runx2-positive cells in the PTB was significantly reduced in SM-EV-treated mice (Fig. 9e, f). Osteoclast numbers per field, identified as multinucleated TRAP-positive cells adjacent to bone appeared also reduced, although the difference did not reach statistical significance (Supplementary Fig. 8c).

We also examined the possibility that SM-EVs may affect the differentiation into osteoclasts. Unlike osteoblasts, differentiation of osteoclasts from mouse BM cells was not affected by treatment with SM-EVs (Supplementary Fig. 8d). Thus, altogether, the results are consistent with the conclusion that SM-EVs can execute alterations in bone morphometry in vivo and reduce trabecular bone density, primarily by hindering osteoblast differentiation and maturation with no detectable effects on osteoclast differentiation.

Pearson's correlations within our cohort of patients with SM and osteoporosis/osteopenia where $T$-scores were documented ($n = 13$) showed a significant correlation between $T$-scores and the levels of miR-23a in EVs ($p = 0.05$). Although the correlation between $T$-scores and levels of miR-30a in EVs were not statistically significant, there was a trend to higher values with lower $T$-scores, which we found supportive, given that $T$-scores develop over time and are influenced by other factors including diet and activity (Supplementary Fig. 9a). In a related analysis, both the levels of miR-23a and miR-30a significantly correlated with the levels of total alkaline phosphatase in serum, which together with other indications throughout the paper (Fig. 8a, b and Supplementary Figs. 3b, 8a, b), are consistent with the mechanism described herein as a contributor to bone disease in patients with mastocytosis.

## Discussion

Bone disease, including osteopenia, osteoporosis, osteolytic lesions, and osteosclerosis, is one of the most frequent complications in patients with SM and is associated with bone pain and pathologic fractures, which are debilitating and cause a significant decline in the quality of life of these patients[3,6,13,37]. It is known that bone disease in SM correlates with mast cell number in the marrow and has generally been attributed to mast cell infiltration and the local release of mast cell-derived products. As in most cases of osteoporosis regardless of the cause, it is understood that a shift in the balance between bone formation and bone resorption with a preponderance of bone resorption must occur. However, the underlying pathophysiology of bone deterioration in SM remains unknown[3]. Here, we implicate low density, small EVs derived from neoplastic mast cells as previously unsuspected regulators of osteogenesis by delivering miRNAs, particularly mir-23a and miR-30a, that restrict key transcriptional osteoblast differentiation programs without apparently affecting osteoclast differentiation. The study provides clues into the pathology of bone disease in SM and offers new possibilities for exploration of other roles for mast cell-derived EVs in osteoporosis and therapies for SM bone disease.

EVs released from bone cells participate in the regulation of bone remodeling and homeostasis[18]. In addition, in diseases such as multiple myeloma, EVs released from cancer cells can cause

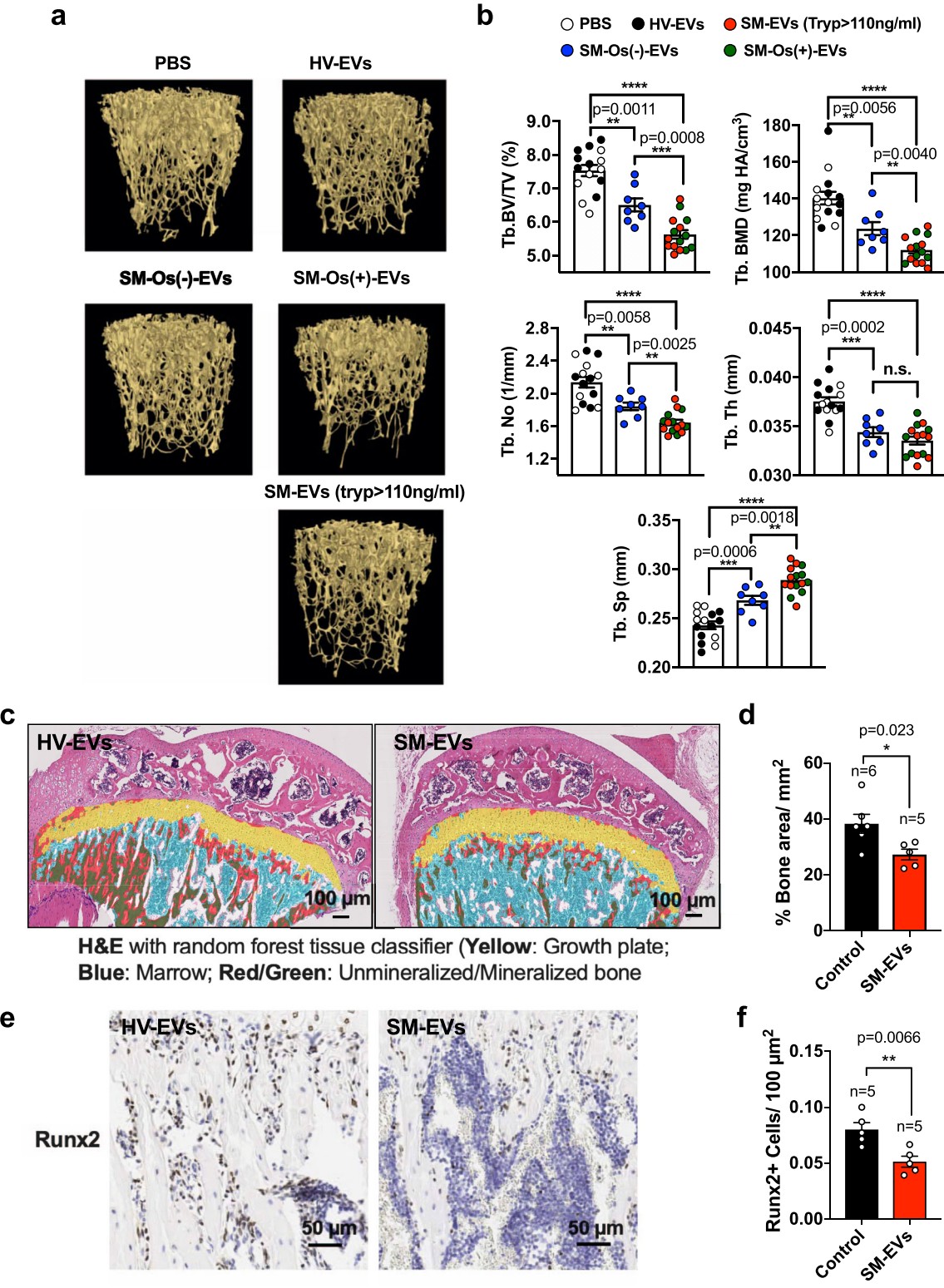

H&E with random forest tissue classifier (**Yellow**: Growth plate;
**Blue**: Marrow; **Red/Green**: Unmineralized/Mineralized bone)

imbalances in such homeostasis and contribute to the development of osteoporosis[38]. Among the cargoes EVs carry, miRNAs are considered key players in the complex regulation of bone coupling[18,19] and in the pathogenesis of bone diseases such as osteoporosis, osteoarthritis, and rheumatoid arthritis[30,39]. A miRNA profiling study in a cohort of patients with osteoporosis identified eleven upregulated miRNAs in blood and/or bone tissue[31]. We found that four of those (miR-21a, miR-23a, miR-93a, and miR-25) were also

increased within SM-EVs. All these miRNAs, in addition to miR-30a (also upregulated in SM-EVs), have been linked to osteoblast differentiation and regulation of cell fate and bone development[31,34], and all were effectively transferred into osteoblasts (Fig. 4b). However, we present evidence that only miR-30a and miR-23a are essential for the inhibition of osteoblast differentiation by SM-EVs: first, as found in other osteoblast cell lines[33,34], transfection of hFOB1.19 with mimics for miR-30a or miR-23a, but not mimics for

**Fig. 9 EVs isolated from patients with SM causes bone alterations in recipient mice. a** Representative μCT 3D images of trabecular (Tb) bone obtained with the Bruker-μCT software suite. Mice were injected with PBS or a pool of extracellular vesicles isolated from the serum of healthy volunteers (HV-EVs) as controls, and compared to mice injected with pooled EVs from all patients with systemic mastocytosis (SM) and tryptase >110 ng/ml (SM-EVs (Tryp > 110 ng/ml)), EVs from patients with SM with osteoporosis/osteopenia (SM-Os(+)-EVs), or patients without osteoporosis/osteopenia (SM-Os(−)-EVs) as indicated. **b** Quantification of percent trabecular (Tb) bone volume per tissue volume (BV/TV), trabecular number (Tb. No.), spacing (Tb. Sp.), and thickness (Tb. Th). No differences were observed in the cortical bone. Empty circles represent mice injected with PBS ($n = 7$ mice); black circles, mice injected with HV-EVs ($n = 8$), blue circles those injected with SM-Os(−)-EVs ($n = 8$), red circles are mice injected with SM-EVs (tryp > 110 ng/ml), and green circles represent mice injected with SM-Os(+)-EVs ($n = 8$), as described in **a**. Data are presented as the mean ± SEM. **$p < 0.01$, ***$p < 0.001$; ****$p < 0.000$.; and n.s. not significant, using an unpaired Student $t$-test (two-tailed). Exact $p$ values when ≥0.0001 are also indicated. **c** H&E-stained histology slides of decalcified proximal tibial bones analyzed using a random forest tissue classifier. Color code is indicated in the figure. **d** Quantification of the fraction of bone per $mm^2$ in the proximal tibial bone images in **c**. **e** Immunohistochemistry staining of Runx2 in decalcified proximal tibias. **f** Quantification of Runx2-positive cells per area of decalcified tibias as shown in **e** by immunohistochemistry. In **c–f**, mice were injected with HV-EVs or SM-EVs as in Fig. 8c ($n = 5$ or 6 mice per group as indicated in the figure). In **d** and **f**, data are the mean ± SEM. *$p < 0.05$; **$p < 0.01$ using an unpaired Student $t$-test (two-tailed).

miR-25, miR-21, or miR-93, inhibited osteoblast differentiation and mineralization, reproducing the attenuating effect of SM-EVs. Second, the inhibitory effects of EVs from neoplastic mast cells on osteoblast differentiation in cultures (Fig. 6d, e) and when injected into mice (Fig. 8e) were reversed after specific knockdown of miR-30a or miR-23a, indicating a significant contribution of each to the overall effects of these EVs. Furthermore, the fact that silencing either miR-23a or miR-30a markedly reversed the effects of SM-EVs on osteoblasts suggests redundancy in their mechanism of action, as will be discussed below. Third, the levels of miR-30a and miR-23a in EVs from individual patient samples significantly correlated with their ability to inhibit osteoblast differentiation in culture (Fig. 4c). Last, the levels of these two miRNAs in EVs derived from patients with SM-osteopenia/osteoporosis were higher than in EVs from non-osteoporotic/osteopenic patients (Supplementary Fig. 3b), caused larger inhibition of osteoblast differentiation/maturation in culture (Fig. 8a), and resulted in drastic reductions in bone formation markers and bone volume when injected into mice (Figs. 8d and 9a, b), confirming a general association with bone pathology. This was also supported by a general association between the levels of miR-23a and $T$-scores in patients. Although other miRNAs with distinct or perhaps redundant activities, or other molecules within EVs in combination with miR-30a and miR-23a may contribute to the effect of SM-EVs, the finding that single knockdown of either miR-30a or miR-23a can prevent such regulation show promise of strong therapeutic potential.

MiRNAs are small RNAs (18–25 nucleotide) that bind to their target mRNAs leading to the silencing of gene expression, either through the inhibition of mRNA translation or by mRNA destabilization. They effectively modulate various cellular processes because a given miRNA may bind to hundreds of mRNA targets and 3′ UTR regions of most mRNAs contain binding sites for multiple miRNAs[40]. Our study also provides insights into the pathways affected by these miRNAs. We find that miR-23a, miR-30a, or SM-EVs reduced expression of RUNX2, a global regulator of osteogenesis[41] and SMAD1/5, which mediates BMP-induced osteogenic function, and SMAD2 but to a much lesser extent. While miR-23a and miR-30a similarly reduced RUNX2 expression, miR-30a was most effective in reducing SMAD1, and miR-23a in reducing SMAD5. This relative specificity was predicted by up to five different computational models that determine potential miR targets, including Diana tools, Miranda, Mirbridge, Targetscan, and Pictar (compared using miRSystem[36] http://mirsystem.cgm.ntu.edu.tw). In other osteoblast models, both miR-23a and miR-30a were demonstrated to target RUNX2[33–35], and miR-30 family members to target SMAD1[34]. In addition, the miR cluster 23a-27a-24-2 was shown to target SATB2, another transcriptional activator of osteogenesis[33], although we did not find an effect for SM-EVs, miR-23a, or miR-30a on SATB2 expression. SMAD1/5/8 are activated by most BMPs, a group of

factors of the TGF-beta superfamily that induce maturation of osteoblasts and are produced in an autocrine manner by mesenchymal osteoprogenitor cells[42]. In addition, BMPs enhance the expression of RUNX2 which, in turn, induces BMP expression[42]. As BMP-mediated signals and RUNX2 synergize in the induction of osteoblast maturation programs[42], our finding of miR-23a, miR-30a affecting both BMP signaling and RUNX2 suggest that SM-EVs, through these miRNAs, intercept two critical synergizing components resulting in effective disruption of osteoblast maturation.

RUNX2 mRNA expression was also reduced in the BM cells of patients with bone disease and in mouse BM cells after repetitive injection of mice with EVs derived from neoplastic mast cells or SM-EVs, particularly those isolated from patients with bone disease, demonstrating their potential in the execution of osteogenesis disruption. Furthermore, SM-EVs markedly altered bone architecture when injected into mice, reducing bone volume and the number of trabeculae without increasing osteoclast numbers or affecting osteoclast differentiation in vitro. In the literature, there is some support for the notion of impaired bone formation in SM. For instance, the levels of the bone formation marker osteocalcin were reduced in patients with SM-related osteoporosis compared to patients with no osteoporosis, although no differences in bone-ALP activity were observed between these groups[43]. Similarly, we did not find differences in ALP mRNA expression in BM in SM with osteoporosis/osteopenia compared to those without (Fig. 8b) despite reduced RUNX2 levels. Our data on the mRNA expression of RANK, RANKL, and OPG in BM cells of SM with osteoporosis/osteopenia suggest a complex regulation of bone metabolism but do not predictably implicate osteoclastogenesis since the ratio of RANKL/OPG mRNA was reduced in these patients compared to those without bone disease (Supplementary Fig. 8a), suggesting bone remodeling as others have reported using additional bone turnover markers[2–4]. Given that the regulation of osteoblast and osteoclast differentiation and function is tightly orchestrated, the expression of these markers may not necessarily shed light into the pathogenesis, as they could reflect compensatory mechanisms. However, the data presented here are consistent with the conclusion that EVs shed by neoplastic mast cells control expression of RUNX2 to attenuate osteoblastogenesis and bone formation in these patients.

In summary, the study implicates for the first time small, low density EVs from neoplastic mast cells in the skeletal abnormalities in patients with SM by delivering miRNAs that inhibit key osteogenic transcriptional programs thus controlling bone production and contributing to bone disease. The identification of miR-23a and miR-30a in SM-EVs does not exclude other networks of players in the shifting of bone coupling in SM but adds another dimension to the understanding of osteoporosis in SM.

Based on our results, further investigation on how other contents of mast cell-shed EVs (related to age, gender, or disease severity) may influence the bone environment are warranted. Caution also must be taken in applying these studies to more aggressive disease variants where expansion of other cell types, such as occurs in mastocytosis with as associated hematologic neoplasm, will lead to mixed sources of EVs, and/or where treatment with more aggressive therapies such as steroids, 2-CDA, alpha interferon, and KIT inhibitors may well alter the number, composition, and biologic effects of EVs. This being said, based on the mechanisms described herein for mast cell-derived EVs, manipulation of EV contents such as antisense for single miRNAs for delivery in vivo or regulation of EV shedding[17,44,45] may constitute new avenues worth considering in future therapeutic approaches.

## Methods

**Patients of study**. Twenty-one adult patients with ISM, diagnosed in accordance with the World Health Organization (WHO) 2008 guidelines[46], were enrolled and consented on protocol NIAID 02-I-0277 approved by the National Institute of Allergy and Infectious Diseases (NIAID) Institutional Review Board in agreement with the Declaration of Helsinki; and which was structured to provide information to address disease pathogenesis and prognosis. To be entered into the protocol, participants had to be between 2 and 80 years of age, be under the care of a primary care physician, have histologic evidence of increased mast cell number by BM and/ or skin biopsy and undergo a BM biopsy and aspirate. Participants did have the option to decline a BM aspirate and biopsy. Exclusionary criteria were a hemoglobin <8 g/dl or a hematocrit <24. Samples from de-identified healthy nonallergic volunteers were obtained and consented through the NIH Department of Transfusion Medicine under NIH protocol 99-CC-0168. All participants provided written informed consent. Sample collection and the studies using these samples were conducted in compliance with the ethical regulations determined on the approved protocols by the NIH Institutional Review Board.

The cohort of patients in this study included principally patients with the indolent form of mastocytosis (ISM) and was a similar cohort to that examined in a previous report[22]. When indicated, samples from additional patients were used (Supplementary Table 1). Patients with ISM constitute the largest group of patients with mastocytosis (>90% in most studies), generally live a normal life span, and because of the long course of disease, experience significant skeletal problems. Data from such patients are less likely to be confounded by associated hematologic disorders or need for cytoreductive therapy[47]. Stored samples from patients with mastocytosis (generally referred to as SM in this report), obtained under our natural history protocol 02-I-0277, were randomly selected to include a wide range of tryptase values in blood, a surrogate marker of mast cell burden[46,47]. They were then divided into two subgroups in correlation with mast cell burden: those with tryptase values below and those above the median tryptase level (110 ng/ml)[22]. The percentage of patients diagnosed with osteoporosis or osteopenia was 40% in the group with tryptase <110 ng/ml and 80% in the group with tryptase >110 ng/ml. Osteoporosis and osteopenia were diagnosed using a dual energy X-ray absorptiometry scan. According to WHO guidelines, patients with a $T$-score < −2.5 were recognized as having osteoporosis, and those with a $T$-score between −1.0 and −2.5 as osteopenia, at either the femoral neck, total femur, or AP spine. None of these patients were under treatment with bisphosphonates or corticosteroids at the time of the collection of the samples and only one patient was on Gleevec.

**EV isolation and characterization**. EVs from individual serum samples (100 μL) or BM supernatants were purified using ExoQuick solution[48,49] (System Bioscience) according to the manufacturer's instructions. EVs from each group of subjects (HV, SM with tryptase <110 ng/ml, and SM with tryptase >110 ng/ml) were, in most cases, pooled and diluted with PBS before testing. Protein content in the EV preparations was determined and 10–100 μg of the individual samples or sample pools (as indicated) were used for studies on target cells. The average EV number and size distribution were determined by NTA using a NanoSight NS300 system (NanoSight). As we previously reported[22], EVs isolated by ExoQuick precipitation had similar characteristics of size and appearance (determined by electron microscopy and NTA), and content (determined by western blot) to those isolated by ultracentrifugation. ExoQuick-ULTRA methods were also used as directed by the manufacturer to obtain further enriched preparations of EVs (with reduced contents of serum proteins) with identical results to Exoquick.

When indicated, EVs were isolated by density gradient fractionation[27]. Briefly, serum was diluted ~1:20 in PBS and centrifuged at 15,000 × $g$ for 40 min and the supernatant filtered through a 0.22-μm Millipore filter. Clarified supernatants were ultracentrifuged at 120,000 × $g$ for 4 h to pellet small EVs. Pellets of small EVs (P120) were washed twice with cold PBS, and further purified by high-resolution iodixanol (Sigma-Aldrich, St Louis, MO, USA) density gradients fractionation (12%–36%) by resuspending the EVs (P120) in 36% iodixanol solution (in PBS) and layering on top of it descending concentrations of iodixanol. The gradients were subjected to ultracentrifugation at 120,000 × $g$ for 15 h at 4 °C. Fractions of 1

ml were collected from the top of the gradient, diluted in PBS, and subjected again to ultracentrifugation at 120,000 × $g$ for 4 h at 4 °C and their effects tested on osteoblast differentiation (see illustration in Supplementary Fig. 1b). In other experiments, the first half or the second half of the gradient fractions were pooled, diluted in PBS, and processed as explained above, and 100 μg of EVs tested for their functionality on osteoblast cultures. Enriched EVs using this method showed effects that appeared indistinguishable to those obtained with EVs isolated by ExoQuick.

For isolation of EVs from BM, aspirates were obtained from patients with SM after informed written consent, a clinical study approved by the Institutional Review Board of the NIAID (02-I-0277) in agreement with the declaration of Helsinki. Supernatants, after centrifugation for 15 min at 490 × $g$, were collected and EVs isolated using Exoquick. These EVs (100 μg) were examined for their effects on hFOB1.19 osteoblast differentiation over the course of 15 days as described below.

EVs released from HMC-1.1 and HMC-1.2 mast cells, or from HMC-1.2 after silencing of miR-23a and miR-30a were isolated using ExoQuick-TC (System Bioscience). Cells (50 × 10^6) were cultured for 3 days in a T75 in 10 mL of media containing 10% EV-depleted FBS (obtained by ultracentrifugation at 100,000 × $g$ for 18 h). The culture media was then collected and EVs isolated with ExoQuick-TC following manufacturer instructions.

**Cell cultures**. The human fetal osteoblastic cell line hFOB1.19, obtained from the American Type Culture Collection (ATCC), was conditionally immortalized with a gene encoding a temperature-sensitive mutant (tsA58) of SV40 large T antigen (LTA)[23]. At the permissive temperature of 33.5 °C, the ts-SV40LTA is active and the hFOB1.19 cells proliferate rapidly, whereas at the non-permissive temperature of 39.5 °C, the ts-SV40LTA is inactive, and the cells differentiate and display the phenotype of mature osteoblasts[23,24]. The cells were cultured in a 1:1 mixture of phenol-free Dulbecco's modified Eagle's medium:Ham's F-12 medium (Invitrogen) supplemented with 10% FBS, 100-U/ml penicillin, 100-μg/mL streptomycin, and 0.3-mg/ml G418 in a humidified atmosphere of 5% $CO_2$. In this study, hFOB1.19 cells were cultured at 39.5 °C to induce differentiation into osteoblasts. Transfections of these cells were also performed at 39.5 °C.

hMSCs media and osteogenic differentiation media components were purchased from Lonza (Walkersville, MD). hMSCs were maintained in DMEM (Lonza) supplemented with 10% fetal bovine serum (FBS), 100-U/mL penicillin, and 100-μg/mL streptomycin at 37 °C under a humidified atmosphere with 5% $CO_2$. For differentiation into bone cells, hMSCs were cultured with the same medium supplemented with 10-mM β-glycerophosphate, 50-μM ascorbic acid, and 100-nM dexamethasone[26].

The human mastocytosis mast cell lines HMC-1.1 (carrying a G560V mutation in one allele of KIT) and HMC-1.2 cells (carrying a G560V and D816V mutation in one allele of KIT)[50] were cultured in Iscove's-modified Dulbecco's medium supplemented with 10% FBS, L-glutamine (2 mM), penicillin (100 U/ml), and streptomycin (100 μg/ml) as described[51]. HEK293 cells were cultured in DMEM supplemented with 10% FBS, L-glutamine (2 mM), penicillin (100 U/ml), and streptomycin (100 μg/ml).

**Effects of miRNA mimics on osteoblast differentiation**. Chemically modified double-stranded RNAs with the sequences of miR-21, miR-23-a, miR-25, miR-30a, and miR-93 or control miRNA (miRNA mimics) were purchased from Qiagen (see Supplementary Table 2 for the sequences). hFOB1.19 cells were seeded and incubated in complete medium without antibiotics for 1 day before transfection with the miRNA mimics (100 nM) or negative control miRNAs using HiPerFect Transfection Reagent (Qiagen). Transfection was performed in antibiotic-free OptiMEM according to the manufacturer's instructions. Two days later, transfection medium was replaced with growth medium and ALP activity released into the media was determined 5 days later. In an additional experiment to test earlier effects, transfection was performed for 1 day and ALP activity determined 1 day later.

**Effects of miRNA inhibitors on osteoblast differentiation in the presence of EVs**. Chemically modified single-stranded RNA molecules that silence specific miRNA species or control miRNAs (Control-miR) (see Supplementary Table 2 for the sequences) were purchased from Qiagen. Osteoblastic cells were seeded as explained above and treated with EVs (100 μg) 2 h later and every 2 days with every change of media for a total of 6 days in culture. Cells were transfected with 100 nM of miRNA inhibitors or negative control after incubation in medium without antibiotics for 1 day as explained above for the miRNA mimics. The transfection media were removed 2 days later and fresh growth medium added. ALP activity in the media and cellular RUNX2 expression were assessed 5 days later.

For transfection of HMC-1.2 with miR-23a and miR-30a inhibitors, cells were seeded onto 24-well plates (7 × 10^4 cells/well) and transfected with the inhibitors or negative control miRNA (100 nM) after 1 day in antibiotic-free media. After 2 days, the media were replaced by fresh media. Media were collected 3 days later and EVs isolated.

**Measurements of miRNA or RNA in cells, BM, or EVs.** MiRNA species in EVs or in cells were determined by qRT-PCR. RNA was extracted using a RNeasy Plus Mini kit (Qiagen) from the following types of samples: EVs (100 μg, isolated from patient serum or the culture media of HMC-1.2 cells); hFOB1.19 treated with SM-EVs; hFOB1.19 cells transfected or not with miRNA mimics; or hFOB1.19 treated with SM-EVs and transfected with miRNA antagonists as explained above. When isolating RNA from 100 μg EVs (corresponding to ~10–50 μl of serum), $1.6 \times 10^8$ copies/μL of a spike-in control mi-RNA (miR-39) was added as an extraction/normalization control. Extracted RNA (100 ng) was reverse-transcribed using miScript II RT Kit (Qiagen) in a 20 μL reaction mixture according to the manufacturer's instructions. After cDNA synthesis of mature miRNAs, qRT-PCR was performed using the miScript SYBR® Green PCR Kit (Qiagen) together with specific miRNA primers for each of the species measured (miRScript Primer Assay, listed in Supplementary Table 3) in a CFX96 Real Time PCR Detection instrument and BioRad CFX Manager software (BioRad, Hercules, CA).

Cells (hFOB1.19 or hMSC) were plated and treated with a pool of EVs from HV, SM with tryptase <110 ng/ml or SM with tryptase >110 ng/ml for up to 12 days or transfected with RNA mimics as described above. At the indicated times, total cellular RNA was extracted from $5 \times 10^5$ cells using a RNeasy Plus Mini kit (Qiagen). RNeasy Plus Mini kit (Qiagen) was also used to extract RNA from BM cells flushed from the femurs and tibias of mice (treated or not with EVs), calvaria's osteoblasts, or BM mononuclear cells from patient's biopsies. RNA (1 μg) extracted in this manner was converted to cDNA by reverse transcription using random hexamers and SuperScript III reverse transcriptase (Invitrogen Life Technologies) in a 20-μl reaction mixture according to the manufacturer's instructions. Quantitative RT-PCR for RUNX2, SATB2, SMAD1/2/5, RANK, RANKL, OPG, and TRAP was performed using a SYBR® Green PCR master mix (Qiagen) together with the specific primer sequences. GAPDH was used for normalization. The specific primer sequences are listed in Supplementary Table 3.

**Osteoblast differentiation and mineralization assays.** hFOB1.19 cells were plated in 12-well plates ($2 \times 10^5$ cells/well) and 2 h later treated with EVs from the indicated groups (100 μg). Culture media were replaced every 3 days and EVs were replenished with every media change. Osteoblast differentiation at the indicated times was assessed by measuring the levels of ALP activity; RUNX2, COL1, and OPN expression by Western blot (see list of antibodies in Supplementary Table 4); and mineralization by alizarin red staining as explained below.

ALP activity in cell culture supernatants was determined using an Alkaline Phosphatase Diethanolamine Activity Kit (Sigma). The supernatants of cultured cells (100 μL) were incubated with 400 μL of reaction buffer (1.0-M diethanolamine and 0.5-mM $MgCl_2$, pH 9.8) in the presence of 50 μL of 0.67 M $p$-nitrophenyl phosphate (ALP substrate) for 30 min at 37 °C. The reaction was stopped with 50 μL of 1-N NaOH, and the hydrolyzed substrate detected by absorbance at 405 nm using a plate reader (Spectra max 340PC, Molecular Devices), with a SoftMax Pro 6.2 software. ALP activity was calculated in units. One unit was defined as the amount of ALP that hydrolyzes 1 μmol of substrate per min at 37 °C. The activity of ALP found in the culture media was corrected by the amount of cellular protein in each well. Lysates were obtained by solubilization of cells in lysis buffer (150-mM NaCl, 10-mM Tris-HCl (pH 8.0), 1-mM EDTA, 1-mM $Na_3VO_4$, 0.5-mM PMSF, 5-mg/mL aprotinin, 5-mg/mL leupeptin, complete protease inhibitor cocktail (Roche) and 1% NP-40) for 15 min and protein content was determined with a Bradford Assay (Bio-Rad).

Maturation of hFOB1.19 into bone-producing osteoblasts was confirmed by alizarin red staining. At the indicated times, culture media were removed, and cells were fixed in 4% paraformaldehyde for 20 min, washed two times with PBS, and then stained with 1% alizarin red solution (Sigma-Aldrich) for 30 min. After washing five times with PBS to eliminate excess dye in the wells, 500 μl of PBS was added and cells examined for the presence of calcium deposits. The red dye staining the calcium deposits was then extracted by shaking the plates for 60 min at room temperature in PBS. The supernatants were then transferred to clean dishes and the absorbance of the extracted dye was measured at 562 nm using a plate reader (Spectra max 340PC).

**Western blotting.** Human FOB1.19 cells ($2 \times 10^5$ cells in 1 mL of culture media) were plated in 12-well plates and treated with EVs as described above. At the indicated times, cells were washed thoroughly and lysed in lysis buffer for 15 min. Protein content in the lysates was determined using a Bradford Assay and equal amounts of protein were loaded into SDS-PAGE gels. Proteins were transferred into nitrocellulose membranes. Membranes were rinsed and blocked with 5% BSA in Tris-buffered saline for 1 h, and then incubated with the specific primary antibodies (Supplementary Table 4) overnight at 4 °C[22]. Immunoreactive bands were detected using infrared dye-conjugated secondary antibodies conjugated (LI-COR Biosciences) (1:20,000) (Supplementary Table 4) and imaged and quantified using a LI-COR Odyssey imager (LI-COR Biosciences) and Image Studio (v5.2) software. Uncropped blots and markers are shown in the Source Data file.

**Isolation of osteoblasts from mouse calvaria.** Mice treated with EVs, as illustrated in Fig. 8c, were euthanized and calvarial bones were dissected. After removal of the periosteum, blood vessels, and interstitial cartilage, calvarial bones were washed with Hanks' solution, excised into 1–2-mm³ fragments and digested sequentially in 0.25% trypsin containing 0.02% EDTA for 25 min (at 37 °C) to eliminate fibrous tissue; and then in 0.1% (w/v) collagenase I and 0.05% trypsin containing 0.004% EDTA (in Hanks' solution) for 1 h (at 37 °C) in an orbital shaker (Eppendorf Thermo Mixer) at 200 rpm. Cells were collected by centrifugation at $218 \times g$ for 8 min and plated in 25-cm² tissue culture flasks in 5 ml of α-MEM containing 1-g/L D-Glucose, L-Glutamine and 10% (vol/vol) of heat inactivated FBS. Fibroblasts were allowed to adhere for 20 min at 37 °C in a 5% $CO_2$ atmosphere, and the rest of the cells (which did not adhere in this time frame) were subcultured into another flask (this step was repeated twice). Non-adherent cells were allowed to attach and cultured to 80% confluency, with change in culture medium after 3 days. Cells were used 6 days after plating for RNA analysis.

**Osteoclast differentiation from mouse BM.** Mouse BM cells were isolated from the femurs and tibias of mice (7 weeks old).

The cells were cultured in α-MEM containing 10% FBS for 24 h. Non-adherent cells were then cultured in α-MEM containing 10% FBS, 2-mM L-glutamine, and 100-U/mL penicillin–streptomycin with 30 ng/ml M-CSF for 3 days. After 3 days, the attached cells were used as bone marrow macrophages (BMMs). For differentiation of osteoclasts, BMMs ($1 \times 10^4$ cells/well in 96-well cluster plate) were incubated in α-MEM supplemented with 10% FBS, 2-mM L-glutamine, 100-U/ml penicillin–streptomycin, 30 ng/ml of M-CSF, and 100 ng/ml of RANKL for 4 days with or without 100 μg of EVs from the indicated groups. Cells were fixed with 4% paraformaldehyde for 10 min and permeabilized with 0.5% Triton X-100 in PBS for 5 min. The supernatants were removed, and the cells incubated with TRAP substrate (SIGMAFAST™ p-Nitrophenyl phosphate; Sigma-Aldrich) solubilized in TRAP buffer (12 g/L of sodium tartrate in 120-mM sodium acetate buffer; pH 5.2). After 30 min, the solution was transferred to a 96-well plate and absorbance measured at 405 nm as a measure of total TRAP activity per well. The cells were then stained for tartrate-resistant acid phosphatase (TRAP) using a leukocyte acid phosphatase kit (Sigma-Aldrich; Cat#387-A), washed three times with $dH_2O$ and examined under a microscope. TRAP+ cells containing more than three nuclei were considered osteoclasts.

**Analysis of bone in mice treated with SM-EVs.** Mice were housed in AAALAC-accredited NIAID animal facilities, with the temperature at 20–25 °C and humidity at 30–70%, a 12-h light–dark cycle and ad libitum access to NIH 31 feed and hyper chlorinated water. Experimental protocols with live mice were performed under an animal study proposal (LAD2E) approved by the NIAID-DIR-Animal Care and Use Committee, in accordance with federal regulatory requirements and ethical standards under the guidance of the Office of Animal Care and Use of the National Institutes of Health.

EVs obtained from serum of HV or patients with SM from the indicated groups were pooled and injected into C57/BL6 female mice (5-week-old) via the tail vein once a day every 2 days for a total of nine injections (200 μg of EVs in 100 μL of sterile PBS per mouse). Alternatively, mice were injected or not with 200 μg of EVs isolated from the cell culture supernatant of HMC-1.2 cells transfected with control miR antagonists or antagonist oligos for miR-23a, miR-30a, or both. Mice were euthanized 2 days after the last injection. RNA was extracted from mice femurs and tibias, or osteoblasts isolated from calvaria for determination of RUNX2, ALP mRNA, and other transcripts as described above. Hind legs of individual mice were placed in 10% buffered formalin and kept for μCT or histomorphometry as described in Supplementary Methods.

For the analysis of bone by high-resolution micro X-ray computed tomography, fixed femurs of mice treated with EVs as described above were scanned using a Skyscan1172 (Bruker microCT, Kontich, Belgium) at a nominal resolution (pixel size) of 4.87 μm, with the x-ray source (focal spot size, 4 μm, energy range 20–100 kV) biased at 45-kV/222 μAmps and using an aluminum 0.5-mm filter to reduce beam hardening. Five hundred projections were acquired with an angular resolution of 0.4° through 180° rotation. Five frames were averaged for each projection radiograph with an exposure time of 725 ms per frame. Tomographic images were reconstructed using SkyScan™ NRecon based on the Feldkamp[2] cone beam algorithm. Ring artifact reduction of 6 and beam hardening correction of 30% were applied.

Trabecular and cortical bone regions of the femurs were defined as segments along the long axis of the distal femur relative to the growth plate reference level. A SkyScan CT-Analyzer software suite was used for 3D model reconstructions and bone morphometric analyses, including trabecular bone volume fraction (BV/TV), trabecular thickness (Tb. Th), number (Tb. No), spacing (Tb. Sp), trabecular bone density (BMD), and cortical volume. Trabecular measurements were taken 0.78 mm from the growth plate reference and extended for 2.5 mm of trabecular structure. Cortical measurements were taken 3.65 mm from the growth plate reference and extended for 1 mm.

**Immunohistochemistry.** Hind leg bones from mice euthanized 2–3 weeks after injections with EVs were obtained and cleaned from soft tissue and fixed in 10% neutralized buffered formalin. Bones were decalcified in EDTA and embedded in paraffin using standard procedures. Sections of paraffin embedded bone (5 μm) were mounted on positively charged slides, air dried, and then baked at 60 °C for

1 h in an incubator. Slides were deparaffinized, rehydrated with alcohol gradients, and placed into Antigen Retrieval Citrate buffer (Vector Labs, CA, USA) for 17 h in a 60 °C water bath. Sections were rinsed in dH$_2$O, loaded onto Leica Biosystems Bond RX autostainers, and stained with anti-Runx2 (Abcam# ab192256) diluted 1:1000. Bond Polymer Refine Detection Kit (Leica Biosystems) was used to detect the immunocomplexes with omission of the Post Primary Reagent. An isotype control was used as a negative control to demonstrate lack of nonspecific binding of anti-Runx2. The number of Runx2-positive cells per μm² square was determined using HALO analytical algorithm. TRAP stain was used to identify osteoclasts in red with blue nuclei. Acid phosphatase staining was used as a control. TRAP-positive multinucleated osteoclasts were counted manually in five fields at 40× objective lens. Slides were also stained with H&E following standard protocols. H&E-stained bone slides were scanned using Aperio AT2 scanner (Leica Biosystems, Buffalo Grove, IL) into whole slide digital images. Image analysis of H&E-stained slides was performed using Halo imaging analysis software (v3.1.1076.423; Indica Labs, Corrales, NM) by one pathologist (B.K.) who was blinded to the status of the groups of mice. Image annotations were performed at the level of growth plate including metaphysis and upper portion of diaphysis. Random forest tissue classifier was used as the algorithm to automatically differentiate bone areas (red and green depicting mineralized and unmineralized, respectively) in H&E slides. In addition, histopathological examination was performed under microscope for H&E slides.

**Statistical analysis**. Each experiment was performed with 3–5 biological replicates, or as indicated in the legends to figures. Experiments were reproduced minimally three independent times with similar results. All statistical tests were performed using GraphPad Prism 8 (v 8.1.1) (GraphPad Software Inc.). Statistically significant differences were calculated by using the Student $t$-test (unpaired, two-tailed) or Mann–Whitney test (two-tailed) as indicated in the figures. Two-way ANOVA was used for comparisons of curves. No multiple comparisons were performed.

**Reporting summary**. Further information on research design is available in the Nature Research Reporting Summary linked to this article.

## Data availability

All data supporting the findings of this study are available within the main manuscript and the supplementary files or provided upon reasonable request. Raw data and original gel images corresponding to all the main and supplementary figures are included in the Source Data file. Source data are provided with this paper.

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

## Acknowledgements

We thank Robin Eisch in her role as protocol study coordinator, Linda Scott, the nurse practitioner on the study protocols, and Yun Bai for aid maintaining stored samples. We also thank the NIH Mouse Imaging Facility for supporting high-resolution micro-computed tomography (μCT). This work was supported by the Division of Intramural Research of the National Institutes of Health (NIH), National Institute of Allergy and Infectious Diseases (NIAID).

## Author contributions

D.-K.K. performed the majority of the experimental work in this study, analyzed data, and contributed to the writing of the manuscript; G.B., Y.-E.C., M.-C.B., and H.M.K. performed experiments and analyzed data; H.D.K. obtained BM aspirates from patients, and provided samples and information of all patients; D.R.D. assisted with μCt and bone analysis; B.K. performed the bone histomorphological analysis. D.D.M. supervised the study and wrote the manuscript; A.O. designed and supervised the study, interpreted and analyzed data, and wrote the manuscript. All authors critically revised and approved the manuscript.

## Funding

## Competing interests

The authors declare no competing interests.
