## [Peer Review File · Nature Communications]

Reviewers' Comments:

Reviewer #1:

Remarks to the Author:

-

Reviewer #2:

Remarks to the Author:

Previously, the authors showed that mastocytosis patients have increased serum levels of extracellular vesicles (EVs) and that these EVs can transfer RNA and protein to other cells, influencing phenotype. In this paper, the authors show that EVs from the serum of patients with mastocytosis have increased levels of certain miRNAs that have an inhibitory effect on osteoblastic differentiation in vitro. They also show that injection of mice with EVs from patients causes a decrease in trabecular bone volume that is accompanied by a decrease in mRNA for selected osteoblastic genes, such as Runx2, in flushed bone marrow. These studies are of interest because osteopenia/osteoporosis is frequently associated with mastocytosis, and the mast cell product, histamine, can have an inhibitory effect on osteoblastogenesis, with a stimulatory effect on osteoclasts. This novel work provides a potential mechanism by which mast cells/mast cell products can influence the bone microenvironment.

For the most part, the studies appear to be rigorously performed, particularly the in vitro work. Effects of EVs and EV-miRNAs on osteoblastic differentiation are carefully demonstrated in a human cell line as well as human mesenchymal stem cell cultures. Weaknesses in the work stem from the in vivo studies; primarily what is missing from the data. Specific comments include:

1. The patient and healthy control populations need to be better described, including demographics such as age, gender and ethnicity. Comments as to how well the healthy control subjects were matched with the patients are needed.
2. Regarding the alizarin red staining for hFOB cells in Figure 2b, the staining in the high magnification images for the "EV-" samples doesn't seem to match that of the whole well. Specifically, staining of the whole well seems much more intense. This is a minor concern.
3. The in vivo studies appear to be performed with N=2 animals per group. While it is appreciated that these studies are difficult and expensive to perform, additional subjects are needed to satisfy rigor and reproducibility. Moreover, if/when more animals are treated, the authors should perform dynamic bone labeling studies, so that bone formation rate may be quantified. If the author's hypothesis is correct, then bone formation may be decreased in mice treated with patient EVs. Further, if more animals are treated, the authors should consider isolating RNA from mouse calvaria. This tissue can be a good source for osteoblast/osteocyte enriched RNA. Excise the calvaria at euthanasia, immediately freeze in liquid nitrogen and homogenize in trizol.
4. The authors show only microCT data for the in vivo results. Missing are histomorphometry data on these samples. Quantifying osteoblast and osteoclast number and surface will provide critical information on the mechanisms driving changes in bone volume (more osteoclasts and/or fewer osteoblasts?).
5. In the introduction, the authors briefly discuss RANKL/OPG ratios in patients with mastocytosis. The authors should examine the ratios of these RNAs in bone and/or bone marrow.

Reviewer #3:

Remarks to the Author:

In the manuscript entitled "Mastocytosis-derived extracellular vesicles deliver miR-23a and miR-30a into pre-osteoblasts and prevent osteoblastogenesis and bone formation" by Do-Kyun Kim et al., authors provide novel and interesting in-vitro, and in-vivo evidences of the role mastocytosis-

derived EVs in the prevention of osteoblastogenesis. In addition, authors identified miR-23a and miR-30a as the molecular mediators of these effects. Authors show that miR-23a and miR-30a affect osteoblastogenesis by targeting osteoblast markers, such as RUNX-2 and SMAD. In my opinion the manuscript can be accepted for publication in Nature Communication following major revisions.

Major revisions:

1. EV characterization was performed by EV size distribution. In line with the MISEV 2018 guidelines in addition to these analyses, morphological characterization (SEM or TEM or AFM) and biochemical analysis (western blot for specific EV markers) should be included in the manuscript. Furthermore, authors should isolate EVs by ultracentrifugation with density gradient and test which of the fractions has the functional activity, see Jeppens D.K's paper in Cell 177, 428.445, 2019.
2. I firmly believe that the paper should contain data also regarding the effects of mastocytosis-derived extracellular vesicles on osteoclastogenesis. As the authors correctly state, bone formation depends on the balance between bone formation and bone degradation, in order to be publishable in Nature Communications, experiments on macrophage differentiation in osteoclast should be performed.
3. Mostly of the experiments have been conducted with patient-isolated EVs, and this clearly reinforces the experimental evidence. At this point my doubt is what is the source of the serum from which the vesicles are isolated? As understood from the manuscript, the authors isolate vesicles from peripheral blood. However, as reported by authors, systemic mastocytosis is characterized by the abnormal accumulation of mast cells in the bone marrow; in fact, one of the criteria for the diagnosis of systemic mastocytosis is the presence of mastocyte aggregates in the bone marrow. Therefore, although the use of EV from peripheral blood is still an added value, I believe that some of the experiments should be conducted using EV from the bone marrow aspirates in order to more specifically isolate mast cells-EV.
4. Other osteoblast markers, other than ALP, should be used: Collagen I and OPN
5. Since the key role of BMP pathway in osteoblastogenesis and since the authors checked the activation of SMAD pathway that is downstream of BMP, authors should check if there is a SM-EV mediated downregulation of BMPs.
6. In order to clarify the role of miR23a and miR30a on RUNX2, and SMAD expression, I suggest evaluating their expression also at early time point, 24h after transfection.
7. In line 172, authors assess that the treatment with SM-EVs increase levels of miRNAs in hFOB1.19 for at least 3 days, but what about at early time point? Usually EV uptake by target cells occurs quite quickly and miRNA is quickly processed; therefore, earlier time points should be added.
8. In Figure 2B the treatment with HV-EVs is missing and this is in contrast with lines 133-134.
9. In Figure 2D western blot of total Akt and total Erk is missing.
10. In order to correlate the effects of SM-EV observed in mice with the miRNAs previously analysed, in vivo experiments should be implemented by the injection of miRNA mimics and/or of SM-EVs and miRNA inhibitors.
11. Statistical analysis: which kind of un-paired T test was conducted? A two-tailed un-paired T test is required.

Minor revision

1. From line 52 to line 103 they should specify that is the section of "Introduction" or "background".
2. In lines 165-166 they refer to a miRNA PCR analysis not shown, they could add this data in supplementary.
3. After utilizing 2 different neoplastic mast cell line, in line 242 they assess that only HMC-1.2 were transfected with miRNA inhibitors; is there a reason why they choose this cell line instead of transfecting the other one?
4. In the scheme in Figure 3a the EV treatment is missing.
5. The concept explained in line 263 should be clarified, avoiding the use of symbols (">, =").
6. In all wb add the molecular weight of the detected proteins.

We thank the referees for their careful evaluation of the paper and positive comments. We believe that we have successfully fulfilled the requests and concerns of the reviewers; and which have greatly enhanced the quality, novelty and impact of the manuscript.

Reviewers' comments:

Reviewer #1 (Remarks to the Author):

Referee 1 (with clinical expertise in systemic mastocytosis) did not leave an official report, but confidentially stated that the numbers of patients is not sufficient to correlate EVs and miRNAs with severity of bone disease, especially with bone density = T Score, noting that the disease is heterogeneous and also patients are further divided into subgroups.

By far the majority of patients with mastocytosis fall within the category of indolent systemic mastocytosis (greater than 90% in most series) which is the patient population which is the principal focus of this paper (see Supplementary Table 1). Patients with indolent systemic mastocytosis generally live a normal life span, and thus suffer greatly from the long-term effects of the increased mast cell burden on the skeletal system. As to the other categories, isolated cutaneous disease is unusual in adults. The remainder of patients fall within the aggressive mastocytosis categories where there are often additional mutations, an associated hematologic neoplasm and requirement for cytoreductive agents, all of which confound any studies on the effect of mast cells on the skeletal system. Thus, we believe this study on the effects of mast cells on the skeletal system focusing mostly on indolent systemic mastocytosis is appropriate and the number of patients sufficient, particularly noting the incorporation now of additional patients into the study. A sentence clarifying the population studied is added to Pg. 13 in the methods section and in Supplementary Table 1.

As to the second concern, in this study we have not determined correlations between EV miRNAs and bone scores in patients and thus it is unclear what this comment specifically refers to. In part, we interpreted that reviewer 1, similar to the other reviewers, is suggesting to increase the number of mice injected with EVs from patients. Fig. 9 is the only figure where micro-Ct-scans and bone density were presented. However, this is not in patients but in mice. Mice were injected with either of two separate pools of EVs (from at least 10 patients each) for a uniform representation of the function of EVs. In the revised manuscript (new Fig. 9) we have injected 5-6 additional mice (now a total of 8 to 15 animals per group) with SM-EV pools to increase the confidence for statistical significance, which was achieved also with smaller animal numbers in our previous version (maximum of 4 mice per group). The data confirms that EVs from patients with SM have the ability to cause marked changes in trabecular bone number, thickness, spacing and density.

In addition, and perhaps also relating to this reviewer's concerns, we have now increased the number of patients relating to:

- 1- Measurements of the expression of RUNX2, ALP and other transcripts such as RANKL, OPG (also requested by reviewer 2), RANK and TRAP in bone marrow cells from patients. Now we present data in new Fig. 8b and Supplementary Fig. 7 a and b (see answer 5 to reviewer 2) on 12 patients with SM but no osteoporosis/penia and 20 patients with SM and osteoporosis/penia (Supplementary Table 1c) instead of the 8 and 8 (respectively) originally presented in Fig 8b.
- 2- Correlations between EV-miRNA content and effect of EVs on the inhibition on osteoblast differentiation in vitro (Fig. 4c). We have now collected EVs from 10 additional individual patients, measured their relative contents in miR-21, miR-23a, miR-25, miR-30a, and miR-93 (in new Supplementary Fig. 2b), and correlated them with the effects of these EVs on osteoblast differentiation (New Fig. 4c, open circles). This separate set of experiments with a different population of patients, confirms and maintains (now in a total of 31 patients) the correlation between the content of miR-23a, and 30a (and not the other miRNAs) and the inhibition of osteoblast differentiation by SM-EVs.

Reviewer #2 (Remarks to the Author):

Previously, the authors showed that mastocytosis patients have increased serum levels of extracellular vesicles (EVs) and that these EVs can transfer RNA and protein to other cells, influencing phenotype. In this paper, the authors show that EVs from the serum of patients with mastocytosis have increased levels of certain miRNAs that have an inhibitory effect on osteoblastic differentiation in vitro. They also show that injection of mice with EVs from patients causes a decrease in trabecular bone volume that is accompanied by a decrease in mRNA for selected osteoblastic genes, such as Runx2, in flushed bone marrow. These studies are of interest because osteopenia/osteoporosis is frequently associated with mastocytosis, and the mast cell product, histamine, can have an inhibitory effect on osteoblastogenesis, with a stimulatory effect on osteoclasts. This novel work provides a potential mechanism by which mast cells/mast cell products can influence the bone microenvironment.

For the most part, the studies appear to be rigorously performed, particularly the in vitro work. Effects of EVs and EV-miRNAs on osteoblastic differentiation are carefully demonstrated in a human cell line as well as human mesenchymal stem cell cultures. Weaknesses in the work stem from the in vivo studies; primarily what is missing from the data. Specific comments include:

We thank this reviewer for the positive comments.

1. The patient and healthy control populations need to be better described, including demographics such as age, gender and ethnicity. Comments as to how well the healthy control subjects were matched with the patients are needed.

Details on the patient populations including demographics are now included in new Supplementary Table 1(a-c). The NIH blood-bank provided adult de-identified normal healthy volunteers who are non-allergic, with a similar age median (48 years old) but detailed

demographic data for privacy concerns is not made available. Information that is available is included in Supplementary Table 1 legend.

2. Regarding the alizarin red staining for hFOB cells in Figure 2b, the staining in the high magnification images for the “EV-“ samples doesn't seem to match that of the whole well. Specifically, staining of the whole well seems much more intense. This is a minor concern.

The two types of images should not exactly match as they don't represent the same aspects. The magnified images in Fig 2b (now the top panel) are pictures of cells in the center of 12 well plates after staining the calcium deposits with alizarin red and extensively washing the excess dye. The picture was intended to illustrate: 1) the presence of extracellular calcium nodules characteristically deposited by mature osteoblasts; and 2) that the treatment with EVs does not affect the number of cells or confluency of osteoblast cultures (we also provide cell numbers in the Results section). The dye trapped in the extracellular deposits (shown in this figure) was extracted for 1 h, and a macroscopic picture of the wells was then taken (now the lower panel), showing both the red supernatants of homogeneously diffused, extracted dye, and the remaining dye in the nodules. Thus, the microscopic view is shown on top and the macroscopic on the bottom. Note that Fig. 2b has been modified (as requested by reviewer 3) to include also the results of osteoblasts treated with HV-EVs which were not shown in the previous version. The supernatants containing extracted dye were then transferred to a clean well and absorbance measured (bar graph in the bottom of Fig. 2b). We believe we have now clarified this data with additions to the results (Pg.4 and 5), legend of Fig. 2b, and supplementary methods.

3. The in vivo studies appear to be performed with N=2 animals per group. While it is appreciated that these studies are difficult and expensive to perform, additional subjects are needed to satisfy rigor and reproducibility. Moreover, if/when more animals are treated, the authors should perform dynamic bone labeling studies, so that bone formation rate may be quantified. If the author's hypothesis is correct, then bone formation may be decreased in mice treated with patient EVs. Further, if more animals are treated, the authors should consider isolating RNA from mouse calvaria. This tissue can be a good source for osteoblast/osteocyte enriched RNA. Excise the calvaria at euthanasia, immediately freeze in liquid nitrogen and homogenize in trizol.

As requested by this reviewer as well as by other reviewers, we now have injected additional mice for micro-Ct measurements of bone and the results are now presented in new Fig. 9 in combination with the mice of the previous experiment. The new data confirms the previous observations but with greater statistical significance that long bones of mice injected with EVs from patients with SM show reductions in bone trabecular number, thickness and trabecular bone volume and density compared to mice injected with HV-SM.

In addition, as requested by the reviewer, we obtained RNA from osteoblasts isolated from the calvarial bones of these mice, and measured mRNA for ALP and RUNX2 by q-RT PCR. The results confirm our previous data that SM-EVs pools, particularly SM-EVs from patients with osteoporosis, when injected into mice, significantly reduce the expression of these transcripts. The data are now shown in Fig. 8d, side by side with the data in bone marrow from the hind legs, as indicated, and described in Pg. 9. We however, were not able to do dynamic bone labeling

studies, but do believe that the combination of micro-Ct, histomorphometric data (see below) and RNA expression in bone marrow cells is compelling and consistent with an effect in osteoblast differentiation and bone formation.

4. The authors show only microCT data for the in vivo results. Missing are histomorphometry data on these samples. Quantifying osteoblast and osteoclast number and surface will provide critical information on the mechanisms driving changes in bone volume (more osteoclasts and/or fewer osteoblasts?).

As suggested by the reviewer, we analyzed decalcified bones by histology (in new Fig. 9 and Supplementary Fig. 7c). Histomorphometric data indicate a reduction in bone spicules and significantly lower numbers of Runx2 positive osteoblasts (New Fig. 9 c-f); and no differences in osteoclast staining, although there was a trend towards lower (but not higher) numbers of osteoclasts (Supplementary Fig. 7c). Also, related to effects on osteoclasts, we tested the effects of SM-EVs in the differentiation of osteoclasts in culture (this was requested by reviewer 3; see response to question 2) and found no differences (see Supplementary Fig. 7d). These new results are detailed on Pg. 9, second paragraph.

5. In the introduction, the authors briefly discuss RANKL/OPG ratios in patients with mastocytosis. The authors should examine the ratios of these RNAs in bone and/or bone marrow.

Following this reviewer's suggestion, we have now determined the ratio of RANKL/OPG in mRNA from bone marrow cells in aspirates from patients with mastocytosis. We found a significant reduction in the ratio of RANKL/OPG in patients with mastocytosis and osteoporosis/osteopenia versus those without (Supplementary Fig.7a). This was due to a slight increase in OPG mRNA expression, accompanied by reductions in RANKL mRNA in the bone marrow cells. While the increased OPG in mastocytosis with osteoporosis/penia agrees with the reported increase in OPG protein levels in the serum of patients with ISM compared to normal subjects (Rabenhorst et al; J Allergy Clin Immunol 132:1234, 2013), the RANKL mRNA levels do not match the increased RANKL protein levels observed in the serum of patients with ISM compared to normal individuals (Rabenhorst et al; J.Allergy Clin Immunol 132:1234, 2013). In the report by Rabenhorst et al, the ratio RANKL/OPG (judging by the reported mean values of RANKL and OPG in serum) appeared to be either unchanged or slightly increased compared to control individuals, but to the best of our knowledge, the ratio RANKL/OPG in mastocytosis with osteoporosis/osteopenia compared with mastocytosis without osteoporosis/osteopenia has not been reported (Rossini et al 2014, J. Allergy Clin Immunol 133:933, 2014). It is important to note that our RANKL/OPG ratio measurements are at the transcriptional level and from bone marrow cells and thus may not completely represent the overall regulatory environment. Nevertheless, the reduced ratio RANKL/OPG mRNA, together with reduced expression of RANK (expressed in osteoclasts) and a trend towards reduced TRAP (a marker of osteoclasts) do not support an increase in osteoclast numbers. In contrast, the reduced RUNX2 and RANKL are consistent with our conclusion of an alteration in osteoblast differentiation. These data are now described in Pg. 9 of the results, discussed in Pg. 12 and shown in Fig. 8b and Supplementary Fig. 7. As an additional note, in increasing the number of samples for measuring RNA transcripts, we found no differences in ALP in the marrows of SM with osteoporosis/penia compared to SM without bone disease (Fig. 8b), suggesting a complex regulatory environment in

vivo (discussed in Pg. 12) where the regulation of ALP by RUNX2 can be overcome by other signals, as ALP can be regulated by also by RUNX2-independent signals (Matsubara et al, J. Biological Chem. 283:29119-29125, 2008; Ding et al. Life Sci, 84:499-504,2009).

Reviewer #3 (Remarks to the Author):

In the manuscript entitled “Mastocytosis-derived extracellular vesicles deliver miR-23a and miR-30a into pre-osteoblasts and prevent osteoblastogenesis and bone formation” by Do-Kyun Kim et al., authors provide novel and interesting in-vitro, and in-vivo evidences of the role mastocytosis-derived EVs in the prevention of osteoblastogenesis. In addition, authors identified miR-23a and miR-30a as the molecular mediators of these effects. Authors show that miR-23a and miR-30a affect osteoblastogenesis by targeting osteoblast markers, such as RUNX-2 and SMAD.

In my opinion the manuscript can be accepted for publication in Nature Communication following major revisions.

Major revisions:

1. EV characterization was performed by EV size distribution. In line with the MISEV 2018 guidelines in addition to these analyses, morphological characterization (SEM or TEM or AFM) and biochemical analysis (western blot for specific EV markers) should be included in the manuscript. Furthermore, authors should isolate EVs by ultracentrifugation with density gradient and test which of the fractions has the functional activity, see Jeppens D.K's paper in Cell 177, 428.445, 2019.

In our recent publication (Kim et al. PNAS 115:E10692-701, 2018), we characterized more extensively the SM-EVs from the same patient population, including biochemical characterization and TEM. TEM showed EV sizes (of about 90-100 nm), consistent with the Nanosight analysis (in Fig. 1b of this manuscript; and Fig. 2 and Supplemental Fig S1 of PNAS). In agreement also with our previous study SM-EV contained markers of small EVs (such as Alix, CD9) similar to HV-EVs but were enriched in characteristic proteins expressed in mast cells such as MRGPRX2. This is shown in part in new Fig. 1c.

In our previous study, we also compared the SM-EVs isolated with Exoquick with those isolated by Exoquick plus additional elimination of serum proteins (with affinity columns), as well as by ultracentrifugation and obtained very similar results in size, biochemical contents, and functionality (Kim et al. PNAS 115:E10692-701, 2018). To satisfy the reviewer's query, we have now isolated SM-EVs by ultracentrifugation in a density gradient according to the protocol in Jeppesen et al, Cell 177, 428.445,2019. A pool of the fractions with lower density (<24% iodixanol gradient), identified as small EVs in Jeppesen et al, and a pool of fractions of higher density (>24% iodixanol gradient), identified as non-vesicle pool, were tested for their effects on osteoblast differentiation. The results show that only the SEV fractions, containing CD9, CD63, Alix and Annexin 1, but not the non-vesicular fractions effectively reduce osteoblast differentiation (Supplementary Fig. 1c). We further tested individual lower density fractions to identify the active fraction, which appeared to be fractions 3-5, at 18-24% iodixanol, coinciding with the maximal expression of Alix, Annexin 1 and the tetraspanins characteristic of small EVs.

The results are now shown in Supplementary Fig. 1b-d, and explained in the results section (Pg. 5, last paragraph).

2. I firmly believe that the paper should contain data also regarding the effects of mastocytosis-derived extracellular vesicles on osteoclastogenesis. As the authors correctly state, bone formation depends on the balance between bone formation and bone degradation, in order to be publishable in Nature Communications, experiments on macrophage differentiation in osteoclast should be performed.

We have performed several additional experiments to address the reviewer's request on the effects of SM-EVs on osteoclasts: 1) We quantified the numbers of TRAP+ cells (osteoclasts) in decalcified femurs of mice (also suggested by reviewer 3) after injection of SM- EVs or HV- EVs. The number of TRAP+ cells did not statistically change in bones of mice injected with SM-EVs compared to controls, although there was a trend towards lower numbers of TRAP positive cells (shown in new Supplementary Fig.7c and described in Pg. 10). 2) We measured the expression of mRNA for RANK and TRAP, expressed by osteoclasts, in bone marrow cells of patients with SM with or without osteoporosis/osteopenia. These transcripts were reduced in SM- osteoporosis/osteopenia compared to SM without osteoporosis/osteopenia (shown in supplementary Fig. 7b and described in Pg. 9). And, 3) We tested the effect of HV-EVs or SM-EV on the differentiation of osteoclasts in vitro from mouse bone marrow hematopoietic stem cells, but found no differences in the number of differentiated osteoclasts (shown in Supplementary Fig. 7d and described in Pg.10). Altogether, we found no evidence for a role of SM-EVs in the differentiation or function of osteoclasts that could explain the reduced bone density in mice injected with SM-EVs. This of course does not exclude a role for osteoclastogenesis in mastocytosis by mechanisms that might be independent of SM-EVs.

3. Most of the experiments have been conducted with patient-isolated EVs, and this clearly reinforces the experimental evidence. At this point my doubt is what is the source of the serum from which the vesicles are isolated? As understood from the manuscript, the authors isolate vesicles from peripheral blood. However, as reported by authors, systemic mastocytosis is characterized by the abnormal accumulation of mast cells in the bone marrow; in fact, one of the criteria for the diagnosis of systemic mastocytosis is the presence of mastocyte aggregates in the bone marrow. Therefore, although the use of EV from peripheral blood is still an added value, I believe that some of the experiments should be conducted using EV from the bone marrow aspirates in order to more specifically isolate mast cells-EV.

One of the criteria for systemic mastocytosis is indeed the presence of mast cell aggregates in the bone marrow, but mast cells accumulate in many other tissues. We previously reported increased EVs with typical mast cell markers in the serum of patients with SM and that the number of EVs in these patients increases with mast cell burden, thus indicating an increased proportion of mast cell-derived EVs in the serum of these patients compared to healthy volunteers (Kim et al. PNAS 115:E10692-701, 2018). Also, our finding that EVs derived from cultured neoplastic mast cell lines phenocopy the effects of SM-EVs on osteoblasts, strongly suggest that mast cell-derived EVs within the population of SM-EVs have the capability of carrying on these actions. Thus, even if the EVs are isolated from serum, this should not detract from the conclusion that EVs derived from mast cells in patients have the potential to alter

osteoblastogenesis. This being said, to address the reviewer's query, we have now isolated EVs from fresh bone marrow aspirates from 3 patients (the only fresh aspirates available at this time; please note that fresh bone marrow aspirates are only done if recommended clinically according to our protocols) and tested their effects on osteoblast differentiation. The data demonstrate that osteoblast differentiation can be inhibited by BM-EVs from mastocytosis patients similar to the inhibition observed with serum SM-EVs (New supplementary Fig. 1a). The results are explained in Pg.5 of the Results section.

4. Other osteoblast markers, other than ALP, should be used: Collagen I and OPN

As suggested by the reviewer, Collagen I and OPN changes in expression by western blot in addition to AP and RUNX2 are now shown in Fig. 2c and d, and explained in Pg. 4 of the manuscript. The data show that treatment of osteoblasts with SM-EVs reduce COL1 and OPN expression by these cells similar to RUNX2, overall consistent with an effect on osteoblast differentiation/maturation.

5. *Since the key role of BMP pathway in osteoblastogenesis and since the authors checked the activation of SMAD pathway that is downstream of BMP, authors should check if there is a SM-EV mediated downregulation of BMPs.*

As suggested by the reviewer, we tested the expression of BMP2 in osteoblasts treated or not with mastocytosis-derived EVs and show that BMP2 expression is reduced, as expected, given the reduction in RUNX2 expression. This is now shown in new Fig. 7b and explained in the results section (Pg. 8) and discussed on Pg. 12, first paragraph.

6. *In order to clarify the role of miR23a and miR30a on RUNX2, and SMAD expression, I suggest evaluating their expression also at early time point, 24h after transfection.*

We have now measured RUNX2 and SMAD expression 24 h after transfection with a 24 h transfection period (2 days total) to ensure the introduction and action of the miRNA mimics, in comparison to the 48 h transfection period and 5 d post-transfection period originally shown in our previous version. The previous and new results (both in new Fig. 7c) on RUNX2 and SMAD expression show very similar patterns, although the reduction in most SMADs (except perhaps SMAD5) is less pronounced 2 d than 7 d after transfection (Fig. 7c), while the effects on RUNX2 are identical. The results suggest that RUNX2 and SMAD5 are primary targets for these miRNAs. Given that miR-30a and 23a preferentially downregulated SMAD1 and 5 respectively (in accordance also with the algorithmic prediction of their respective targets) but similarly downregulated RUNX2, the results suggest that both RUNX2 and SMADs are targets for these miRNAs, although a RUNX2 depletion may also contribute to the overall effect on SMADs. The results are now described on Pg. 8.

7. *In line 172, authors assess that the treatment with SM-EVs increase levels of miRNAs in hFOB1.19 for at least 3 days, but what about at early time point? Usually EV uptake by target cells occurs quite quickly and miRNA is quickly processed; therefore, earlier time points should be added.*

The reviewer is correct that EV uptake can be detected in some cells within 6 h, although maximal uptake and processing of the cargoes may depend on the cells and cargo. As requested, we have now repeated these experiments measuring miRNAs in FOB1.19 osteoblasts including 6, 12 and 24 h after adding EVs. As shown in Supplementary Fig. 2a and noted on Pg. 6, increases in most miRNAs can be observed by 12-24 h, but for miR21 and miR25, modest but significant increases were found by 6h.

8. *In Figure 2B the treatment with HV-EVs is missing and this is in contrast with lines 133-134.*

Osteoblasts treated with HV-EV showed no differences compared to untreated cells and this is why they were not included in all the sets of Alizarin Red experiments. We have now done additional experiments that include the HV-EV treated osteoblast and the results are now shown in Fig. 2b.

9. *In Figure 2D western blot of total Akt and total Erk is missing.*

We repeated the experiment to add total AKT and ERK loading controls that are now shown in Fig. 2d and demonstrate equal loading as illustrated also by actin loading. We also added HV-EV-treated cells in this figure as requested in the previous question

10. *In order to correlate the effects of SM-EV observed in mice with the miRNAs previously analysed, in vivo experiments should be implemented by the injection of miRNA mimics and/or of SM-EVs and miRNA inhibitors.*

We thank the reviewer for this important comment. In answer to this reviewers' request, we have now injected mice with EVs shed from HMC1.2 mast cells transfected or not with control or miRNA inhibitors (antisense) in a similar regimen used for SM-EVs (see scheme illustrated in New Fig. 8c) as this may be the most straight forward demonstration for a role of these miRNAs in the effects on bone in vivo. Similar to SM-EVs, EVs derived from HMC1.2 transfected with control oligos, when injected into mice caused a reduced expression of RUNX2 and ALP in the calvaria, compared to mice injected with PBS. However, when HMC1.2 were transfected with antagonists for mir-23a or mir-30a, the derived EVs with reduced mir-23a or mir-30a (Fig. 6d) did not cause a reduction in RUNX2 and ALP, particularly when both miRNAs were knockdown simultaneously. This is now shown in new Fig 8e and explained in the result section (Pg. 9-10) and discussion (Pg.11). Altogether, the results demonstrate that HMC.1.2 mast cell-derived EVs not only induce similar inhibition of osteoblast differentiation/maturation as SM-EVs in vitro (Fig. 6e), but also reduce RUNX2 and ALP expression when injected into recipient mice (New Fig. 8e); and implicate miR-23a and 30a in these responses in vitro and in vivo.

11. *Statistical analysis: which kind of un-paired T test was conducted? A two-tailed un-paired T test is required.*

Comparisons of experiments on osteoblasts and mice were done with two-tailed un-paired t-tests. Comparisons that had been done with Anova1 tests were re-tested for two-tailed un-paired

Student-tests. The type of statistical method used is mentioned in the methods sections and in each legends to figures.

Minor revision

1. *From line 52 to line 103 they should specify that is the section of “Introduction” or “background”.*

We followed Nature Communications instructions to avoid “Introduction” as a heading

2. *In lines 165-166 they refer to a miRNA PCR analysis not shown, they could add this data in supplementary.*

The reference to the results of this array was actually unnecessary for this paper since we show the levels of the 5 miRNAs we intended, and thus we have deleted the sentence from the paper referencing the array. A complete analysis of this array is underway for future publications.

3. *After utilizing 2 different neoplastic mast cell line, in line 242 they assess that only HMC-1.2 were transfected with miRNA inhibitors; is there a reason why they choose this cell line instead of transfecting the other one?*

We concentrated on HMC1.2 because they produce more EVs, have more of the miRNAs we tested, and are more potent in inhibiting osteoclast differentiation. In addition, this cell line has the most common mutation in SM (KIT- D816V). Thus, it made more sense to knockdown miR-23a and 30a in this cell line for more definitive results. We have included an additional explanation in the text for the use of the EVs from this cell line (Pg.7)

4. *In the scheme in Figure 3a the EV treatment is missing.*

EV treatment is now included

5. The concept explained in line 263 should be clarified, avoiding the use of symbols (“>, =”).

The symbols for the effects on SMADs have been avoided and the effects clarified in the text (Pg. 8)

6. *In all wb add the molecular weight of the detected proteins.*

This is now shown

Reviewers' Comments:

Reviewer #1:

Remarks to the Author:

The manuscript by Do-Kyun Kim and colleagues, 'Mastocytosis-derived extracellular vesicles deliver miR-23a and miR-30a into pre-1 osteoblasts' describes that extracellular vesicles (EV) released from neoplastic mast cells block osteoblast differentiation in culture and thus the mineralization of bones. The manuscript is well written and the data are clear. However, there are a few points that need to be addressed by the authors:

1. If indeed miR-23a and miR-30a play a role in the evolution of osteopenia and osteoporosis in patients with mast cell disease, we would expect that clinical correlations can be documented. Did the authors correlate miR-23a and miR-30a levels in blood with T scores obtained in patients with mastocytosis? Did they correlate miR-23a and miR-30a levels with alkaline phosphatase levels in patients?

2. Did the authors examine whether miR-23a and miR-30a levels change during specific therapy with KIT inhibitors or during treatment with bisphosphonates?

3. Whereas osteopenia and osteoporosis are frequently detected in patients with indolent mastocytosis, osteosclerosis is also diagnosed commonly. Did the authors examine correlations between osteosclerosis and miRNA expression?

4. The authors did not examine the mechanisms of synthesis and release of miR-23a and miR-30a in neoplastic mast cells in detail. Does IgE receptor activation, stem cell factor, or complement components promote expression and/or release of miR-23a and/or miR-30a in mast cells? What signaling pathways are involved?

5. Patients with mastocytosis can be divided into WHO subsets with distinct prognostic profiles and different prevalence of osteopathy. Did the authors correlate WHO variants with serum levels of miR-23a and miR-30a? Did the authors examine whether miRNA levels decrease during therapy with a strong KIT inhibitor or during corticosteroid therapy?

6. What is the mechanism of uptake of miR-23a and miR-30a by osteoblasts? How are these miRNAs delivered from mast cells to osteoblasts and how much is released to form measurable serum levels?

7. How much extra miR-23a and miR-30a are released from mast cells during an anaphylactic event? Did the authors measure miR-23a and miR-30a levels during anaphylactic episodes in patients with mast cell disease?

Reviewer #2:

Remarks to the Author:

For the most part, the authors have been very responsive to the concerns from the original critique. As a result, the work is stronger, particularly the in vivo mouse studies. However, with the additional information about the patient population used for most of the studies, a new concern is realized. In the original patient population (supplemental Table 1a), the patients with higher tryptase levels have a much greater proportion of females compared with the patients with lower tryptase levels. In some of the figures, the authors show that the higher tryptase patients can have higher number of extracellular vesicles (Figure 1a), and more pronounced impact of signaling molecules (Figure 2d, Figure 7b). Are these differential effects truly due to the phenotype associated with higher tryptase or is it because the patient population has a higher proportion of

females? The authors need to address this issue or at least discuss the caveat that sex difference distribution may contribute to the phenotype.

Not all of the western blot data are quantified and analyzed for statistical significance. While this may be less of a problem with protein data that reinforce findings demonstrated at the level of total RNA, it is problematic with data such as that shown in Figure 7b. Here, the authors show decreased BMP2 protein in a western blot image, but the data are not quantified nor are RNA data provided. Further, these data show a difference in response to vesicles isolated from high vs low tryptase patient pools, which relates back to the concern outlined in the first paragraph of this review.

In general, the authors should not be using the word "transcription" when discussing data demonstrating changes in steady state RNA levels. The steady state RNA levels reflect changes mediated by a combination of transcriptional and post-transcriptional regulation, such as RNA stability. This needs to be carefully addressed throughout the paper.

The authors speak of bone spicules (line 392), however, it seems that they are actually referring to bone trabeculae. The authors should clarify.

Reviewer #3:

Remarks to the Author:

Authors have fully addressed all referee's criticisms and the paper can be accepted for publication.

Point by point answers to Reviewers:

Reviewer #1 (Remarks to the Author):

The manuscript by Do-Kyun Kim and colleagues, 'Mastocytosis-derived extracellular vesicles deliver miR-23a and miR-30a into pre-1 osteoblasts' describes that extracellular vesicles (EV) released from neoplastic mast cells block osteoblast differentiation in culture and thus the mineralization of bones. The manuscript is well written and the data are clear. However, there are a few points that need to be addressed by the authors:

We thank this reviewer for the positive comments that the manuscript is well-written, and the data are clear.

1. If indeed miR-23a and miR-30a play a role in the evolution of osteopenia and osteoporosis in patients with mast cell disease, we would expect that clinical correlations can be documented. Did the authors correlate miR-23a and miR-30a levels in blood with T scores obtained in patients with mastocytosis? Did they correlate miR-23a and miR-30a levels with alkaline phosphatase levels in patients?

We thank this reviewer for this question. We found a significant positive correlation between the relative levels of miR-23a and miR-30a in EVs and levels of total alkaline phosphatase in circulation in patients with SM. These correlations are shown in new Supplementary Fig. 9b.

We also correlated the miRNA levels with the T-scores of patients with osteoporosis/osteopenia. The results showed a statistically significant correlation with miR-23a ($p=0.05$; $n=13$). Although there was not statistically significant correlation between the T-score and the levels of miR-30a in EVs, there was a trend to higher levels of miR-30a with lower T-scores which we found supportive, given that T-scores develop over time and are influenced by other factors including diet and activity. The data on these correlations is shown in new Supplementary Fig. 9a and described in Pg.11, lines 419-428 and noted in Pg.12, lines 477-478.

2. Did the authors examine whether miR-23a and miR-30a levels change during specific therapy with KIT inhibitors or during treatment with bisphosphonates?

Within the patient population reported in this study, only one patient (KIT D816V negative) was on a KIT inhibitor (imatinib), and thus general conclusions cannot be drawn, and the result can be taken simply as anecdotal. This patient with a T-score of -3.0 in the spine had relative levels of 11.34 for miR-23a and 16.34 for miR-30a. These miRNA levels within EVs in this patient on Gleevec are lower than in 3 other patients with similar T-scores (ranging between -2.8 to -3.4) that were not taking a KIT inhibitor. The levels of miR23a in these patients were 21.1, 16.2 and 24.3 (mean \pm SD: 20.53 \pm 4.08, versus 11.34 in the patient with imatinib), and those for miR30a were 25.34, 18.99, and 28.72 (mean \pm SD:25.35 \pm 4.94, versus 16.34 in the patient with imatinib).

None of these patients in this cohort were on bisphosphonates when EVs were isolated . We have included a sentence in the Methods to clarify that none of the patients were taking bisphosphonates and that only one was on Gleevec (Pg. 14 Line 587-588).

3. Whereas osteopenia and osteoporosis are frequently detected in patients with indolent mastocytosis, osteosclerosis is also diagnosed commonly. Did the authors examine correlations between osteosclerosis and miRNA expression?

Osteosclerosis was not diagnosed in our patient population.

4. The authors did not examine the mechanisms of synthesis and release of miR-23a and miR-30a in neoplastic mast cells in detail. Does IgE receptor activation, stem cell factor, or complement components promote expression and/or release of miR-23a and/or miR-30a in mast cells? What signaling pathways are involved?

The reviewer is correct, we did not perform in vitro studies to examine release of EVs from stimulated mast cells, rather focusing on EVs released into the blood of patients with mastocytosis in the usual clinical state. Nor were we able to capture EVs released during systemic reactions in patients with mastocytosis, as none were observed while patients were under our care within the NIH Clinical Center. However, it is known that neoplastic mast cells in vitro constitutively release EVs, which if extrapolated to the in vivo situation would explain the presence of EVs with a mast cell signature in blood and in turn levels which correlate with disease and mast cell burden. The mechanisms of EV release including signaling pathways involved is clearly of interest and one which we share with this reviewer. Such studies are a logical extension of this report but beyond its scope.

5. Patients with mastocytosis can be divided into WHO subsets with distinct prognostic profiles and different prevalence of osteopathy. Did the authors correlate WHO variants with serum levels of miR-23a and miR-30a? Did the authors examine whether miRNA levels decrease during therapy with a strong KIT inhibitor or during corticosteroid therapy?

Most patients with SM within this report are patients with indolent disease and where the increased EVs have a mast cell signature associated with the increased mast cell burden. None were on steroids and only one was on a kit inhibitor (see also response to point two above). This has some advantage in the understanding on how mast cells through EV release can effect bone pathology. First, the vast majority of patients with mastocytosis have indolent systemic mastocytosis, so the research approach used in our research design which focuses on the effect of mast cell EVs applies already to the majority of patients. Secondly, when you consider advanced disease, a major category includes patients with an associated hematologic neoplasm. In such cases the EVs in peripheral blood would be from both mast cells and other neoplastic cells which would confound the study, as the effects of EVs would be less likely connected only to mast cells, which is the focus on this study. Further, the majority of patients with indolent mastocytosis receive symptomatic treatment only, while advanced forms are treated variably with alpha-interferon, steroids, 2-CDA, and midostaurine (KIT inhibitor) with unknown effects on exosome formation, making such a study enormously difficult to interpret.

However, within the cohort of patients from whom we measured miRNA levels in EVs, we had 4 with SSM, 3 of which had osteoporosis and 1 where bone pathology was not documented. The data comparing the levels of miR-23a and miR-30a between ISM and SSM in patients with

confirmed bone disease suggest significant increases in these miRNAs in SSM compared to patients with ISM and confirmed bone disease. These data are now shown in new Supplementary Fig. 3d. Furthermore, EVs from these samples caused >70% inhibition in osteoblast differentiation as expected, given the correlation between inhibition of osteoblast differentiation and miR-23a and 30a levels in EVs (see Fig. 4c). Therefore, the data suggests that the mechanism described in this manuscript may also play a role in more advanced mastocytosis where the mast cell compartment is expanded. This is described in Pg. 6, lines 229-234. We also deliberated on the challenges of examining EVs in advanced forms of disease in the discussion in response to this reviewer's concerns (Pg. 13, lines 540-547).

6. What is the mechanism of uptake of miR-23a and miR-30a by osteoblasts? How are these miRNAs delivered from mast cells to osteoblasts and how much is released to form measurable serum levels?

These miRNAs are associated with EVs released into the extracellular space by mast cells. EVs enable intercellular exchange by transferring the molecules they carry into recipient cells. What makes this type of intercellular communication efficient is that the cargos circulate protected from degradation and that the targeting of EVs to recipient cells is deemed relatively specific considering current literature (via recognition of adhesion molecules, receptors or other molecules). When EVs reach the target cell, EVs may then: 1) remain stably associated with the plasma membrane or dissociate; 2) directly fuse with the plasma membrane to release their contents into the cytosol; or 3) be internalized through distinct endocytic pathways. When endocytosed, EVs may subsequently fuse with the endosomal delimiting membrane, releasing their contents into the cytosol, or be targeted to lysosomes for degradation. There are also indications of fusing of EV with the nuclear membrane of the recipient cells.

The docking mechanisms of SM-EVs specifically onto osteoblasts and entry mechanisms remain to be further investigated. This question deserves a detailed study but is one which substantially diverges from the focus of the current report. We do note, however, that we did demonstrate that the miRNAs associated with SM-EVs are successfully delivered into osteoblasts, as their levels increase in osteoblasts after incubation with SM-EVs (Fig. 4b), and that they mediate inhibition of osteoblastogenesis by interfering with RUNX2 and SMADs (Fig. 7), a mechanism that is specific to those miRNAs, since the effects are reversed by specific antisense oligos and mimicked by transfection of such miRNAs into osteoblasts (Figs. 5, 6e and 8e).

Regarding the levels of miRNAs in serum SM-EVs, the levels of the miRNA species presented in this manuscript are detectable in 100 µg of isolated EVs corresponding to approximately 10 to 50 µl of serum (this information was added to Pg. 5 of Supplementary methods).

7. How much extra miR-23a and miR-30a are released from mast cells during an anaphylactic event? Did the authors measure miR-23a and miR-30a levels during anaphylactic episodes in patients with mast cell disease?

We have not evaluated the levels of miR-23a or 30a during an anaphylactic event since we do not have samples from patients undergoing anaphylaxis, as no such events were observed while

patients were under our care within the NIH Clinical Center (see also response to point 4 above). We do demonstrate that constitutive release of EVs by neoplastic mast cells (a common feature to all transformed cells) is associated with their ability to interfere with the differentiation/maturation of osteoblasts.

Reviewer #2 (Remarks to the Author):

For the most part, the authors have been very responsive to the concerns from the original critique. As a result, the work is stronger, particularly the in vivo mouse studies. However, with the additional information about the patient population used for most of the studies, a new concern is realized.

We thank the reviewer for the positive comments

In the original patient population (supplemental Table 1a), the patients with higher tryptase levels have a much greater proportion of females compared with the patients with lower tryptase levels. In some of the figures, the authors show that the higher tryptase patients can have higher number of extracellular vesicles (Figure 1a), and more pronounced impact of signaling molecules (Figure 2d, Figure 7b). Are these differential effects truly due to the phenotype associated with higher tryptase or is it because the patient population has a higher proportion of females? The authors need to address this issue or at least discuss the caveat that sex difference distribution may contribute to the phenotype.

The reviewer is correct in observing that the patient population with higher tryptase values also has a higher proportion of females than the cohort with lower tryptase values. Although EVs from both groups do reduce RUNX2 and SMAD expression and result in inhibition of osteoblast differentiation/maturation, the effect is more pronounced, particularly in on osteoblast signaling, in the higher tryptase group. Gender thus could potentially be a factor accounting for some of these differences between the low and high tryptase groups after EV treatment, although as we will discuss, gender-related differences in the levels of miR-23a and 30a do not appear to be involved. To address the concern of gender in relationship with miRNA levels, we have now compared the levels of miR-23a and 30a in EVs from female patients compared to male patients in our population. The levels of these miRNAs were similar in females and males: miR-23a levels were 16.0 ± 5.1 (mean \pm SD) for females and 16.4 ± 4.2 (mean \pm SD) for males; and miR-30a levels were 20.0 ± 4.6 (mean \pm SD) for females and 19.3 ± 5.2 (mean \pm SD) for males. These numbers are described in the legend of Supplementary Fig. 3c and shown below in the figure below to reviewer together with the levels of other miRNA tested in this study (miR21, 25 and 93). More importantly, new Supplementary Fig. 3c shows that the higher levels of these miRNAs within the population of patients with osteoporosis/osteopenia compared to patients without are not due to gender since their levels, within the population with bone disease, are very similar between females and males. This is noted in Pg.6 lines 227-229.

Even though the levels of these miRNAs are not different according to gender, we cannot exclude the possibility that differences in other molecules within EVs, related or unrelated to gender, may account for the greater signaling effects in the high tryptase group. Thus, in addition

to new Supplementary Fig. 3c, we have added several sentences within the manuscript stressing that other components within the EVs may have a role (related or unrelated to gender) and that this would need further investigation (see Pg.9, lines 328-332 in the results section; and Pg.12, lines 479-480 and Pg.13-14, lines 540-542 of the discussion).

Figure 1 to reviewer- Comparison in the levels of the indicated miRNAs within SM-EVs (all patients) according to gender.

With regards to number of EVs in circulation in association with tryptase values (Figure 1 a), gender also does not seem to be a factor as there are no apparent differences between males and females of similar tryptase values. This is in agreement with other studies reporting no differences in numbers of vesicles between males and females in human subject populations (Eitan et al. Scientific reports, 2017 PMID: [28465537](https://pubmed.ncbi.nlm.nih.gov/28465537/); in Table 2). We have included a new Supplementary Figure 1 to indicate the concentrations of EVs in females and males according to tryptase values, and referred to in Pg.4, lines 130-131.

Not all of the western blot data are quantified and analyzed for statistical significance. While this may be less of a problem with protein data that reinforce findings demonstrated at the level of total RNA, it is problematic with data such as that shown in Figure 7b. Here, the authors show decreased BMP2 protein in a western blot image, but the data are not quantified nor are RNA data provided. Further, these data show a difference in response to vesicles isolated from high vs low tryptase patient pools, which relates back to the concern outlined in the first paragraph of this review.

As requested by this reviewer, we have now provided a quantification of the blot image in new Fig. 7b (n=4) and in new Fig. 2d (n=4). We have also measured mRNA for BMP2 and the data are provided in new Fig 7a (described in Pg.8, lines 311-314). In terms of a difference in response to vesicles isolated from high vs low tryptase, and as detailed above, we found no differences in the miRNAs described here related to gender. We cannot exclude the possibility that differences in other molecules within EVs may account for the enhanced signaling effects in the high tryptase group (which we have discussed in the text). This is a complex question we hope to explore in the future, now that we better understand the role of miRNAs in bone pathology.

In general, the authors should not be using the word “transcription” when discussing data demonstrating changes in steady state RNA levels. The steady state RNA levels reflect changes mediated by a combination of transcriptional and post-transcriptional regulation, such as RNA stability. This needs to be carefully addressed throughout the paper.

We have revised the terminology throughout the manuscript, substituting transcription by “mRNA levels” or “expression”.

The authors speak of bone spicules (line 392), however, it seems that they are actually referring to bone trabeculae. The authors should clarify.

We changed spicules to bone trabeculae (now line 402).

Reviewer #3 (Remarks to the Author):

Authors have fully addressed all referee's criticisms and the paper can be accepted for publication.

Thank you

Reviewers' Comments:

Reviewer #1:

Remarks to the Author:

The authors have addressed all my questions in satisfactory manner and we have no further comments.

Reviewer #2:

Remarks to the Author:

Thank you. The revisions address the concerns from the previous review.